# Heterochromatin *de novo* formation and maintenance in *Plasmodium falciparum*

**Alba Pérez-Cantero**[1☉], **Oriol Llorà-Batlle**[1☉¤a], **Ingrid Pelaez-Conde**[1,2¤b], **César Martínez-Guardiola**[1,2], **Alfred Cortés**[1,3]*

**1** ISGlobal, Barcelona, Catalonia, Spain, **2** Facultat de Medicina i Ciències de la Salut, Universitat de Barcelona (UB), Barcelona, Catalonia, Spain, **3** ICREA, Barcelona, Catalonia, Spain

☉ These authors contributed equally to this work.
¤a Current address: Single Cell Discoveries B.V., Utrecht, the Netherlands
¤b Current address: Max Planck Institute for Molecular Genetics, Berlin, Germany
* alfred.cortes@isglobal.org

## Abstract

In the malaria parasite *Plasmodium falciparum,* the expression of many genes is regulated by heterochromatin (HC) based on the histone mark tri-methylation of histone H3 lysine 9 (H3K9me3). HC assembly involves three distinct steps: *de novo* nucleation, spreading and maintenance. Nucleation, which consists in formation of HC in a previously euchromatic region, determines the specific genomic locations where HC occurs. This process is not well understood in malaria parasites. Here we investigated the DNA sequence *cis* determinants of HC nucleation in *P. falciparum*, using a screening approach based on integration of fragments from different heterochromatic genes into an euchromatic locus, followed by H3K9me3 chromatin immunoprecipitation (ChIP) analysis. We found that fragments of *var* gene upstream regions nucleated HC efficiently, whereas fragments from the *pfap2-g* upstream region or from the *mspdbl2* locus did not nucleate HC. Fragments from the beginning of the coding sequence (CDS) of *pfap2-g* nucleated HC with low efficiency, as evidenced by nucleation requiring long fragments of ~2 kb and occurring only in a fraction of the parasites. These results demonstrate that the primary DNA sequence is a main determinant of HC nucleation in *P. falciparum*. We also studied HC maintenance at the *pfap2-g* locus, which demonstrated that specific parts of the upstream region, different from the regions competent for HC nucleation, are required for maintenance. Together, our results provide initial insight into how HC is directed to specific loci and maintained in *P. falciparum*.

## Author summary

Proper regulation of gene expression is crucial for the functioning of all organisms. One key mechanism involves modifications in the structure of chromatin, the DNA and proteins complex in which the genome is packed. One specific

**Data availability statement:** The new ChIP-seq data presented in this article was deposited in the GEO database with accession number GSE287562.

**Funding:** This work was supported by grants SAF2016-76190-R, PID2019-107232RB-I00 and PID2022-137863OB-I00 to A.C. from the Spanish Ministry of Science and Innovation (MCIN)/ Agencia Estatal de Investigación (AEI, 10.13039/501100011033), co-funded by the European Regional Development Fund (ERDF, European Union). A.P.-C. was the recipient of a Juan de la Cierva fellowship (FJC2021-047724-I) from the Spanish Ministry of Science, Innovation and Universities. O.L.-B. was supported by a FPU fellowship from the Spanish Ministry of Education, Culture and Sports (FPU014/02456) and C.M.-G. by a fellowship from MCIN/ AEI/10.13039/501100011033 (PREP2022-000842). This research is part of ISGlobal's Program on the Molecular Mechanisms of Malaria, which is partially supported by the Fundación Ramón Areces. We acknowledge support from the grant CEX2023-0001290-S funded by MCIN/AEI/10.13039/501100011033, and support from the Generalitat de Catalunya through the CERCA Program. The funders had no role in study design, data collection and analysis, decision to publish, or preparation of the manuscript.

**Competing interests:** The authors have declared that no competing interests exist.

type of chromatin, named heterochromatin (HC), suppresses gene expression and plays fundamental regulatory roles in the majority of eukaryotic cell types, including malaria parasites. However, how HC is assembled and maintained in *Plasmodium falciparum*, the species responsible for the vast majority of malaria cases and deaths, is not known. Here we investigated the role of the primary DNA sequence in determining where in the genome HC is formed and maintained. We identified sequences that always promote HC formation, even when placed away from their natural genomic location, sequences that only promote HC formation in a subset of the cells, and sequences that never promote HC formation. These findings demonstrate that the primary DNA sequence plays a major role in determining where HC is formed. We also identified specific DNA regions that are crucial for HC maintenance. These results provide new insight into HC-based regulation of gene expression in a deadly human pathogen.

## Introduction

In the nucleus of eukaryotic cells, DNA is packed as chromatin, a nucleoprotein complex formed mainly by DNA and histone proteins. Chromatin modulates the accessibility of the transcription machinery to the DNA, which constitutes one of the main mechanisms for the regulation of gene expression. Chromatin can be broadly classified into euchromatin, which is accessible and permissive to transcription, or heterochromatin (HC), which is less accessible and generally prevents transcription. In addition to its major role in the regulation of gene expression, HC is also important for genome stability, silencing of transposable elements and centromere function, among several other roles. A key property of HC is that it can be inherited by epigenetic mechanisms during cell division, such that different patterns of HC distribution in genetically identical cells are inherited in *cis* across multiple cell generations. This makes HC a truly epigenetic trait [1,2].

HC can be constitutive or facultative. Constitutive HC typically occurs in repeat-rich regions of the genome, such as subtelomeric and pericentromeric repeats, which are in a permanently repressive state. Facultative HC is more dynamic and typically affects gene-rich regions that can be in either an euchromatic or a heterochromatic state in different cells with the same genome [1–3]. While HC occurs in different varieties, its formation always involves three distinct steps: nucleation, spreading and maintenance. Post-translational modification of histone tails, mainly methylation of specific lysine residues, lies at the basis of all steps of HC assembly. The molecular machinery responsible for HC formation includes enzymes that modify histone tails, typically referred to as writers or erasers, and enzymes that recognize these modifications, known as readers. The coupling of writer and reader protein domains, together with protein-protein interactions that mediate the recruitment of other regulatory factors, underlies all steps of HC formation [1,2,4].

The first step for HC assembly is *de novo* nucleation, which consists in converting a region that was previously euchromatic into HC. This step determines at which

specific regions of the genome HC is assembled. Multiple HC nucleation mechanisms have been described, which can be broadly divided between RNA-dependent mechanisms and those based on recognition of specific DNA sequences by DNA binding factors. The former, which involve transcripts generated at the site of HC nucleation to recruit chromatin modifying activities, can be divided in RNAi-dependent or RNAi-independent mechanisms [1,2]. Regarding mechanisms based on DNA binding proteins, examples include ADNP or Pax family transcription factors that recognize motifs within repetitive DNA [5,6], ATF family proteins [7,8] and ATRX or KRAB zinc finger proteins that recognize transposable elements [9,10].

After *de novo* nucleation, HC can spread from the nucleation sites to adjacent regions in a sequence-independent manner. Full spreading, which is slow and may require several cell generations, is restricted by limiting amounts of HC factors and by barrier elements (also known as barrier insulators or boundary elements). The latter prevent HC spreading into regions of the genome that need to be accessible [1,11,12]. Finally, inheritance of HC patterns as an epigenetic trait requires maintenance of HC during cell division. Clonal inheritance of HC in *cis* occurs in a self-templated manner, such that the epigenetic memory is stored in HC itself. The molecular mechanism of inheritance involves distribution of nucleosomes with HC-specific histone marks between the sister chromatids during DNA replication. These nucleosomes, located at the same original genome position, are used as templates to add the same marks in the histones of the newly formed nucleosomes [1,2].

The most conserved type of HC across eukaryotic evolution, from yeast to humans, is based on the histone mark di- or tri-methylation of histone H3 lysine 9 (H3K9me2 or H3K9me3). Constitutive HC at repetitive regions is almost invariably associated with this histone mark, whereas facultative HC affecting protein-coding genes can be based on this mark [13,14] or on methylation of lysine 27 (H3K27me3). In particular, facultative HC associated with development of multicellular organisms is often based on H3K27me3 and Polycomb-group proteins [1,3]. Among the key players for the establishment of H3K9me2/3-based HC are lysine methyl transferases such as Suv39 (Clr4 in yeast), which contain a SET domain that mediates methylation of H3K9 and a chromo-domain that recognizes methylated H3K9. The presence of the two domains in the same protein facilitates positive feedback loops needed for HC nucleation, spreading and maintenance, illustrating the importance of coupling reader and writer domains in HC biology. Another key player for the assembly of H3K9me2/3-based HC is heterochromatin protein 1 (HP1), which binds H3K9me2/3 via its chromo-domain and contributes to HC structure through dimerization and recruitment of many enzymatic activities, including histone deacetylases [1–4,13].

In the malaria parasite *P. falciparum*, both constitutive and facultative HC are based on H3K9me3. Constitutive HC is present in subtelomeric repeats, but not in pericentromeric regions. Facultative HC occupies clonally variant gene (CVG) loci, which are located in subtelomeric regions and in a few chromosome-internal islands [15–19]. At CVGs, chromatin can adopt either a transcriptionally permissive (euchromatin) or repressive (HC) conformation. Once established, both states can be stably propagated for many generations (bistable chromatin), with switches between the two states occurring infrequently [16,20–23]. This chromatin dynamics results in variant expression of CVGs, which play a fundamental role in host-parasite interactions by mediating processes such as immune evasion, solute transport, erythrocyte invasion, erythrocyte remodeling or sexual conversion, among others [19,24,25]. Within an isogenic parasite population, differences in CVG expression generate transcriptional and phenotypic diversity, which constitutes the basis of a 'bet-hedging' adaptive strategy [19,26], where cell heterogeneity increases the chances of survival of the population in front of unexpected environmental changes [27,28]. In addition to stage-independent variation between individual parasites, the distribution of HC changes during transmission stages, whereas it remains constant during all asexual blood stages [15,29–31].

The putative *P. falciparum* H3K9 methyltransferase is the SET domain-containing protein SET3 [18,32], although it has not been characterized in detail. Assembly of H3K9me3-based HC in *P. falciparum* also requires the evolutionarily conserved HP1 [15,17,33,34]. However, how HC forms only at some specific loci in the *P. falciparum* genome, resulting in variant expression of the genes affected, and does not form at other sites, still remains unknown. *P. falciparum* does

not have a functional RNAi pathway [35], which suggests that HC nucleation uses an RNAi-independent mechanism in malaria parasites. Several DNA binding factors of the ApiAP2 family [36] are associated with HC [37,38], but none of them has been directly involved in the formation of HC at specific genomic sites.

Here, we aimed to gain initial insight on the mechanism that directs HC formation to specific parts of the genome in malaria parasites. To test if the primary DNA sequence is a major determinant of *de novo* HC nucleation in *P. falciparum* and to identify putative DNA regions with an intrinsic nucleation capacity, we assessed the ability of fragments of different genes to trigger HC formation when ectopically integrated in an euchromatic region of the genome. Our main analysis used fragments from the *pfap2-g* gene [39], which is located in an internal HC island [17,18]. This gene encodes the master regulator of sexual conversion, *i.e.*, the conversion of asexual blood stage parasites into transmissible sexual forms called gametocytes. We also tested the ability to nucleate HC of fragments from another CVG located in a HC island, *mspdbl2* [17,18,40,41], and from several *var* genes (involved in antigenic variation and pathogenesis), located in large HC domains [42]. Furthermore, we analyzed the role of specific *pfap2-g* sequences in HC maintenance by deleting different parts of this locus.

## Results

### Identification of *pfap2-g* transcription start sites (TSSs)

To guide the design of the experiments to assess HC nucleation, ensuring that fragments containing the *pfap2-g* (PF3D7_1222600) TSSs were included, we first mapped the TSSs of this gene using 5' Rapid Amplification of cDNA Ends (5'RACE). This was performed with the E5 line [39] (a 3D7 subclone with a high sexual conversion rate) and the transgenic E5-PfAP2-G-DD line, which has a destabilization domain appended to PfAP2-G and shows unusually high sexual conversion rates when the stabilizing ligand Shield-1 (Shld1) is added [39,43]. In both parasite lines, we consistently identified two TSSs blocks in the *pfap2-g* upstream region, Block 1 and Block 2, located ~1,900 and ~1,600 bp from the start codon, respectively (Fig 1). Of note, previous genome-wide studies on transcription initiation did not identify any TSSs for *pfap2-g* [44,45], likely because the gene was only expressed in a small fraction of the cells in the parasite lines used.

To validate the predicted TSSs identified by 5'RACE, we performed a reverse transcriptase coupled to quantitative PCR (RT-qPCR) analysis using primers located upstream, between or downstream of the two blocks. This analysis confirmed that essentially no transcripts contained sequences upstream of Block 1 and the majority of transcripts contained sequences located between the two blocks (S1 Fig). The irregular pattern observed in the transcript levels measured at different positions may be explained by the occurrence of different mRNA species or by technical limitations associated with the extreme AT-richness of this region.

### Identification of regions of the *pfap2-g* locus with capacity to nucleate HC *de novo*

To identify specific regions of the *pfap2-g* locus able to nucleate HC *de novo*, we designed a screening approach based on the integration of *pfap2-g* fragments into the PF3D7_1144400 locus [16], which is euchromatic (S2A Fig) and encodes a gene dispensable in asexual blood stages [46]. The resulting transgenic lines were analyzed using H3K9me3 chromatin immunoprecipitation (ChIP) coupled to quantitative PCR (ChIP-qPCR) to evaluate HC levels at the site of integration. The analysis of each transgenic line also included heterochromatic and euchromatic genes used as ChIP-qPCR positive and negative controls, respectively (S2A Fig).

To detect HC formation within the ectopically integrated fragments without interference from the endogenous locus and to prevent unwanted plasmid integration via recombination with the endogenous *pfap2-g* gene, we first deleted the upstream regulatory region and part of the coding sequence (CDS) of the endogenous *pfap2-g* gene (positions -3,522 to +992 bp from the start codon) in the 3D7-A subclone 1.2B. This subclone does not form gametocytes due to a non-sense mutation [47] in *gdv-1*, an upstream activator of *pfap2-g* [48]. The resulting parasite line, which we termed Δ5'ap2-g, was

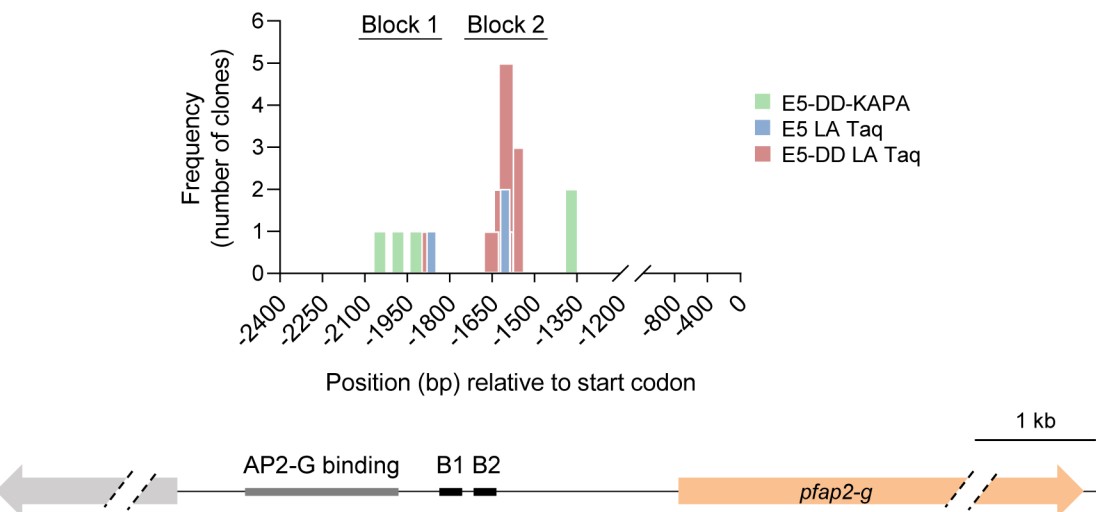

**Fig 1. Identification of *pfap2-g* transcription start sites (TSSs).** Position relative to the start codon and frequency (number of clones) of the TSSs identified by the 5'RACE analysis of *pfap2-g* in the E5 and E5-AP2-G-DD (E5-DD) lines. The E5-DD line was cultured in the presence of Shld1. TSSs were identified from experiments in which PCR was performed with either LA Taq or KAPA DNA polymerases, as indicated. The position of the two main TSS blocks (B1 and B2) and the region containing the previously described PfAP2-G binding motifs is indicated in the schematic (to scale) below.

subsequently edited to integrate the DNA fragments under analysis into the PF3D7_1144400 locus (Fig 2A). The parasite lines used in this study are described in S2B Fig.

We first analyzed by ChIP-qPCR five transgenic lines in which we integrated overlapping ~1 kb fragments (F0, F1, F2, F3 or F4) of the *pfap2-g* upstream region and beginning of the CDS (Fig 2B), which are heterochromatic in the endogenous *pfap2-g* locus [15–18]. The sequence of fragments F1-F4 falls within the region of the endogenous *pfap2-g* locus that was deleted in the Δ5'ap2-g line, whereas part of the F0 sequence was still present at the endogenous locus (Fig 2B). Fragments from the CDS (F0 and F1) and, to a lesser extent, from the proximal upstream region (F2) promoted HC formation, although with a lower coverage than in the ChIP-qPCR positive controls (Figs 2C and S2C). HC was detected within the fragment itself and in the adjacent regions of the PF3D7_1144400 locus, indicating that the newly formed HC was able to spread from the nucleation site into flanking regions. In contrast, fragments from the distal part of the *pfap2-g* upstream region (F3 and F4) did not promote HC formation, as evidenced by similar H3K9me3 levels as in the ChIP-qPCR negative controls (Fig 2C). Additionally, integration of a fragment of similar size from the upstream region of the euchromatic gene *ama1* did not promote HC formation, whereas integration of a fragment from the heterochromatic *var* gene PF3D7_1240300 resulted in prominent HC formation, in this case with a coverage similar to the ChIP-qPCR positive controls (Fig 2B and 2C). The euchromatic or heterochromatic state of the endogenous *ama1* and *var* genes was confirmed by ChIP-qPCR (S2A Fig). To further dissect the sequences that can nucleate HC, we generated additional transgenic lines with shorter fragments of the *pfap2-g* F0 and F1 regions integrated, but clear HC formation was not observed for any of them (S3 Fig). We cannot exclude the possibility that HC was formed in a small fraction of the parasites with these or the F3 or F4 fragments integrated, which would not be detectable in a bulk analysis that reflects the average H3K9me3 occupancy in the population, but this would correspond to highly inefficient nucleation.

Unexpectedly, qPCR and Southern blot analysis of genomic DNA (gDNA) of the transgenic lines revealed integration of multiple copies of the fragment in all cases except for the lines with the *pfap2-g* F4 or *var* gene ~1 kb fragments (Figs 2D and S4) and two of the lines with shorter fragments (S3 Fig). Although the qPCR results are compatible with either integration of concatemers (containing multiple copies of the fragments) or presence of episomal copies of the donor plasmid,

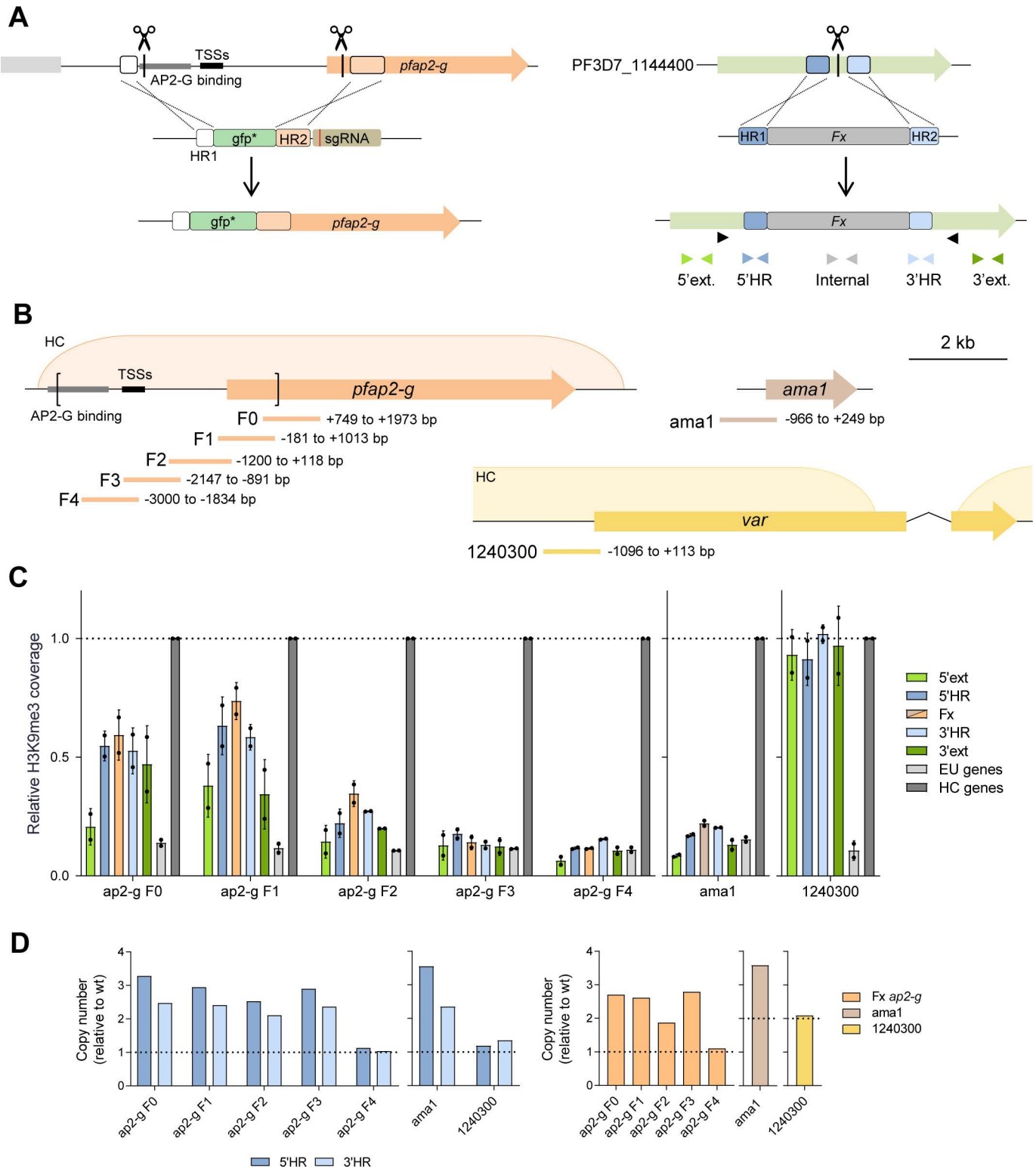

**Fig 2. Identification of regions of the *pfap2-g* locus that can nucleate HC *de novo*.** (A) Left: Strategy for the deletion of the upstream region and part of the CDS of *pfap2-g* using CRISPR/Cas9 (Δ5'ap2-g line). The scissors indicate the positions targeted by single guide RNAs (sgRNAs). Homology regions (HR1 and HR2) are indicated as boxes. A non-functional *gfp* (gfp*) gene was used to replace the deleted region. Right: Strategy for the

integration of different fragments into the euchromatic locus PF3D7_1144400 using CRISPR/Cas9. Fx refers to the fragment to be integrated. Black arrowheads indicate the position of the primers used for diagnostic PCR, and colored arrowheads the position of the primers used for ChIP-qPCR analyses. (B) Schematic (to scale) of the position of the ~1 kb *pfap2-g, ama1* and *var* (PF3D7_1240300) fragments tested for HC nucleation capacity. The transgenic line with the *var* fragment was generated with plasmids containing the *yfcu* marker but, for the experiments presented here, it was analyzed before negative selection, which makes it analogous to the other transgenic lines. Square brackets indicate the region of the endogenous *pfap2-g* locus that was deleted. The previously described position of the HC domains in *pfap2-g* and the *var* gene is indicated (HC) [16]. The position of the TSSs and the region containing the PfAP2-G binding motifs is also indicated. (C) H3K9me3 ChIP-qPCR analysis of the transgenic lines with the integrated fragments (position of primers shown in panel A). Values are the % of the DNA recovered in the H3K9me3 IP (% input) at each position relative to the average % input in the positive control heterochromatic genes *var* PF3D7_1240300 (or *mspdbl2* for the 1240300 transgenic line) and *clag3.2* ("HC genes"). "EU genes" are euchromatic genes actin I (*actI*) and serine-tRNA ligase (*serrs*) used as negative controls (average of the two is shown). The dashed line indicates the coverage in the positive control heterochromatic genes. Data are presented as the average and s.d. of two biological replicates. (D) qPCR analysis of the number of copies of the PF3D7_1144400 HRs (5'HR and 3'HR) and the fragments under analysis in the gDNA of the transgenic lines. Copy number was calculated relative to 1.2B (wild type) gDNA. The dashed lines indicate the expected value for single, correct integration. For fragments of which the endogenous copy was deleted (*i.e.*, *pfap2-g* fragments), the expected value is 1. For fragments of which the endogenous copy is present in the genome, the expected value is 2.

the complex pattern of bands observed in the Southern Blot analysis (S4 Fig) suggests integration of concatemers. While we cannot exclude the possibility that episomal copies of the plasmids were also present and accounted for part of the HC signal detected, HC formation at the F0 and F1 fragments integrated in the genome is demonstrated by the detection of H3K9me3 with the external primer pairs 5'ext. and 3'ext., which recognize sequences adjacent to the site of integration that are not present in the transfection plasmids (Fig 2A and 2C). While H3K9me3 coverage at these positions was lower than in the homology regions or the fragment itself, it was clearly higher than in ChIP-qPCR euchromatic controls. On the other hand, we cannot rule out the possibility that integration of concatemers containing multiple copies of the fragments affects their ability to nucleate HC formation, although the different ability of different fragments to form HC is suggestive of sequence specificity.

### Long fragments are needed for efficient HC nucleation by *pfap2-g* sequences

To characterize the potential effect of integration of multiple copies of the fragments on *de novo* HC nucleation, the *pfap2-g* fragments F0 and F1 were integrated at the PF3D7_1144400 locus using donor plasmids that carried the *yfcu* negative selection marker. This enabled elimination of parasites carrying either episomal plasmid or concatemer integration. ChIP-qPCR analysis of the new transgenic lines containing single copies of fragments F0 or F1 revealed absence of HC at the integrated fragments (Fig 3A and 3B).

  Formation of HC only in parasite lines with multiple copies of F0 or F1 may be explained by the larger total size of putative HC-promoting DNA or by sequence repetitiveness (occurrence of the same sequence more than one time), as repetitiveness itself may promote HC formation. In *Drosophila* and other organisms, presence of multiple copies of a transgene is sufficient to trigger HC-based silencing [49,50]. To distinguish between these possibilities, we assessed HC formation at the PF3D7_1144400 locus after integration of a single copy of longer fragments (~2 kb) from the *pfap2-g* gene (Fig 3C–E). This revealed formation of HC with a single copy of the *pfap2-g* F1 + F0 fragment, including the 5' end of the CDS and a small part of the upstream region, albeit at lower levels than in ChIP-qPCR positive controls (Fig 3D). H3K9me3 signal using primers for the flanking regions (5'ext. and 3'ext.), which was clearly observed, demonstrates that HC was formed at the fragment correctly integrated in the PF3D7_1144400 locus. In contrast, no HC was detected after integration of the F2 + F1 *pfap2-g* fragment or a ~2 kb fragment of the euchromatic *ama1* gene. These results indicate that single copies of fragments of the *pfap2-g* locus can nucleate *de novo* HC formation in a previously euchromatic region, without requiring repetitiveness.

### HC nucleation by *pfap2-g* fragments occurs in only a subset of parasites

H3K9me3 coverage at the HC nucleation-positive *pfap2-g* fragments integrated at the PF3D7_1144400 locus was lower than in ChIP-qPCR positive HC controls (Figs 2C and 3D). To determine whether this reflects that HC is formed in only a

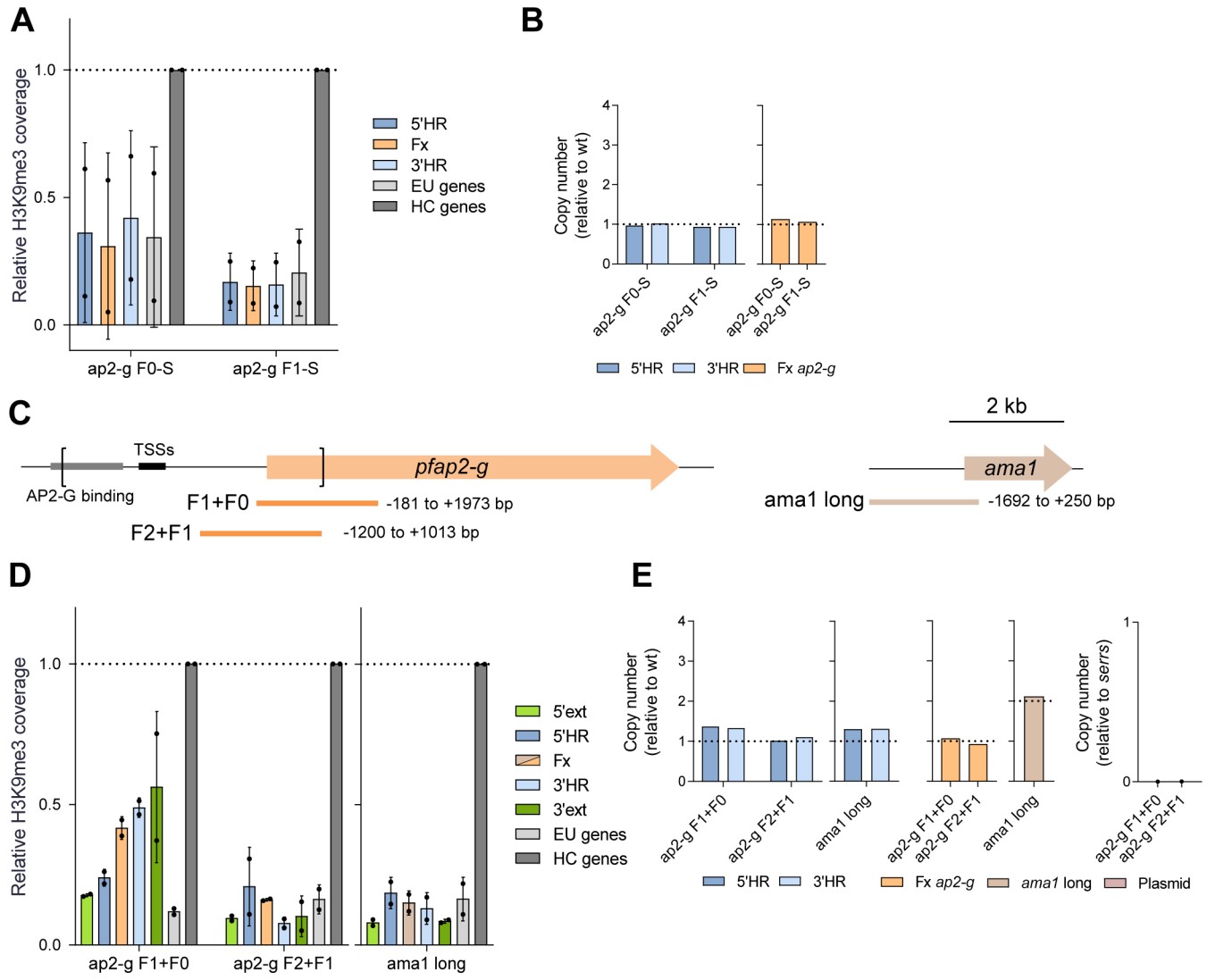

**Fig 3. Effect of fragment size on HC nucleation.** (A) H3K9me3 ChIP-qPCR analysis, as in Fig 2C, of transgenic lines carrying single copies of *pfap2-g* fragments F0 and F1 (ap2-g F0-S and ap2-g F1-S). Data are presented as the average and s.d. of two biological replicates. (B) qPCR analysis of copy number, as in Fig 2D. (C) Schematic (to scale) of the position of the ~2 kb *pfap2-g* and *ama1* fragments. Square brackets indicate the region of the endogenous *pfap2-g* locus that was deleted. The position of the TSSs and the region containing the PfAP2-G binding motifs is indicated. (D) H3K9me3 ChIP-qPCR analysis, as in panel A, of transgenic lines carrying single copies of ~2 kb *pfap2-g* and *ama1* fragments. (E) qPCR analysis of copy number, as in panel B. Additionally, presence of the plasmid backbone was also assessed (using a reference curve constructed with gDNA of the W4-2 parasite line [74], with one integrated plasmid copy).

subset of the parasites, we generated subclones of the transgenic line carrying the *pfap2-g* F1 fragment (multiple copies). In two of the subclones, no HC was observed at the site of fragment integration, whereas in two other subclones (3E and 9B) HC formation was apparent (Fig 4A). HC-positive and -negative subclones contained a similar number of copies of the integrated fragment (Fig 4B). Likewise, analysis of four subclones of the transgenic line with a single copy of the *pfap2-g* F1 + F0 fragment integrated showed that HC formed in two of the subclones (A3 and D3) at levels similar to the ChIP-qPCR positive controls, but was absent in the other two (Fig 4C and 4D). These results confirm that the intermediate levels of H3K9me3 observed in transgenic lines with HC-positive *pfap2-g* fragments reflect population heterogeneity.

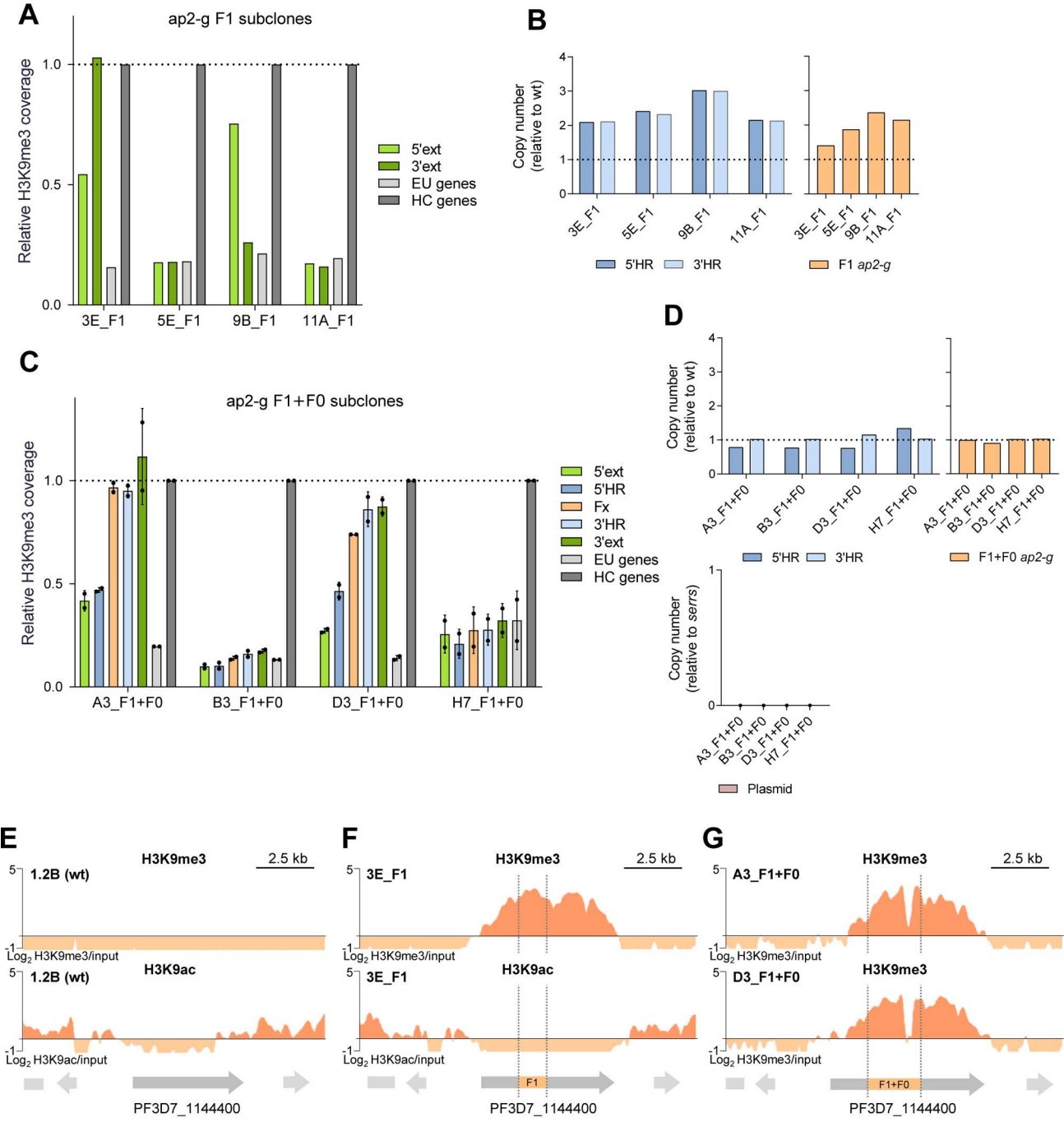

**Fig 4. Cell to cell heterogeneity in HC nucleation by *pfap2-g* fragments.** (A) H3K9me3 ChIP-qPCR analysis, as in Fig 2C, of subclones of the *ap2-g* F1 transgenic line. Data is from a single biological replicate. (B) qPCR analysis of copy number, as in Fig 2D. (C) H3K9me3 ChIP-qPCR analysis, as in panel A, of subclones of the *ap2-g* F1+F0 transgenic line. Data are presented as the average and s.d. of two biological replicates. (D) qPCR analysis of copy number, as in Fig 3E. (E) H3K9me3 (top) and H3K9ac (bottom) ChIP-seq profiles at the PF3D7_1144400 locus in the parental 1.2B line (wild type). Block arrows are annotated genes (start codon to stop codon). Data is from a previously published study [16]. (F) H3K9me3 (top) and H3K9ac (bottom) ChIP-seq profile at the edited PF3D7_1144400 locus in the 3E subclone of the ap2-g F1 transgenic line. Reads were aligned against the edited genome. The vertical dashed lines mark the position of the integrated fragment. (G) H3K9me3 ChIP-seq profile at the edited PF3D7_1144400 locus, as in panel E, in the A3 and D3 subclones of the ap2-g F1+F0 transgenic line.

Additionally, to map the extension of the HC formed at the PF3D7_1144400 locus after integration of nucleation-positive *pfap2-g* fragments, we assessed the distribution of H3K9me3 using ChIP-seq. While HC was absent from the wild-type locus [16] (Fig 4E), in HC-positive subclones with the F1 or the F1+F0 integrated fragments HC spanned over most of the PF3D7_1144400 CDS, such that the limits of the H3K9me3-enriched domain roughly coincided with the beginning and end of the gene (Fig 4F and 4G). In the *pfap2-g* F1 fragment 3E subclone, we also analyzed the distribution of H3K9ac, which occurred at very low levels across the PF3D7_1144400 CDS in wild type parasites [16] (Fig 4E) and was depleted throughout the full heterochromatic region in the 3E subclone (Fig 4F).

### HC formation by sequences from other CVGs

Next, we tested the ability of sequences from other CVGs to nucleate HC ectopically. Following the same approach as for *pfap2-g*, we integrated at the euchromatic locus PF3D7_1144400 fragments from the *mspdbl2* (PF3D7_1036300) locus, which is silenced in the majority of parasites and located in a small internal HC island [16,18,40,41] (S2A Fig), similar to *pfap2-g*, and from three *var* gene loci. These included two subtelomeric *var* genes from group B, PF3D7_0733000 and PF3D7_1041300, and a *var* gene located in a large internal HC island, PF3D7_1240300, from group B/C [51]. We used plasmids containing the *yfcu* negative selection marker to obtain parasites with single copy integrations.

None of the *mspdbl2* fragments tested, including ~1 kb and ~2 kb fragments from the coding and upstream sequences, formed HC at the ectopic site (Fig 5A and 5B). In contrast, all ~1 kb fragments of *var* genes (including upstream regions and in some cases also the beginning of the CDS) robustly formed HC, generally at levels similar to the ChIP-qPCR positive control (Fig 5C and 5D). All transgenic lines with *mspdbl2* or *var* fragments had a single copy of the fragment integrated, with the exception of the line with the PF3D7_1041300 prom+CDS fragment (Fig 5E). However, subclones of this transgenic line with a single copy of the fragment also nucleated HC efficiently, and a small deletion occurring in some parasites with the PF3D7_1240300 fragment did not affect HC nucleation (S5 Fig). Together, these results show that *var* gene sequences can nucleate HC efficiently, whereas some *pfap2-g* sequences can nucleate HC with lower efficiency and *mspdbl2* sequences cannot.

### Analysis of the role of different regions of *pfap2-g* in HC maintenance

The H3K9me3 ChIP-seq analysis of Δ5'ap2-g-derived transgenic lines, in which the upstream region and part of the CDS of the *pfap2-g* locus were deleted (Fig 2A), revealed that HC was absent from the remaining part of the endogenous *pfap2-g* locus (Fig 6A). This unexpected observation indicates that the region deleted is necessary to maintain HC at the full *pfap2-g* locus. In contrast, ChIP-seq analysis of the E5-Δap2-g line, in which most of the *pfap2-g* CDS but not the upstream region was deleted [39], showed that this line maintained HC in the remaining part of the locus (Fig 6A). These results suggest that specific parts of the *pfap2-g* locus are needed for HC maintenance, which prompted us to investigate the determinants of HC maintenance in more detail.

We systematically tested the impact of deleting different regions of the *pfap2-g* locus on the presence of HC at the rest of the locus. The deleted regions were replaced by a non-functional *gfp* gene (gfp*) using CRISPR/Cas9, and HC levels were determined using H3K9me3 ChIP-qPCR with a collection of primers at different positions of the locus (Fig 6B). The ΔCDS transgenic line, in which the deleted region spanned most of the *pfap2-g* CDS, had H3K9me3 levels at the rest of the locus similar to the wild type 1.2B line (Fig 6C and 6D), consistent with the ChIP-seq results for the E5-Δap2-g line (Fig 6A). In contrast, deletion of the *pfap2-g* upstream region (ΔU) resulted in complete HC loss from the full locus (Fig 6D), similar to the Δ5'ap2-g-derived lines (Fig 6A). Together, these results confirm that the *pfap2-g* upstream region but not the CDS is essential for the maintenance of HC.

To narrow down the region needed for HC maintenance, we generated the ΔU5' and ΔU3' lines, in which we removed the distal or the proximal part of the *pfap2-g* upstream region, respectively. While the distal part was not required for HC

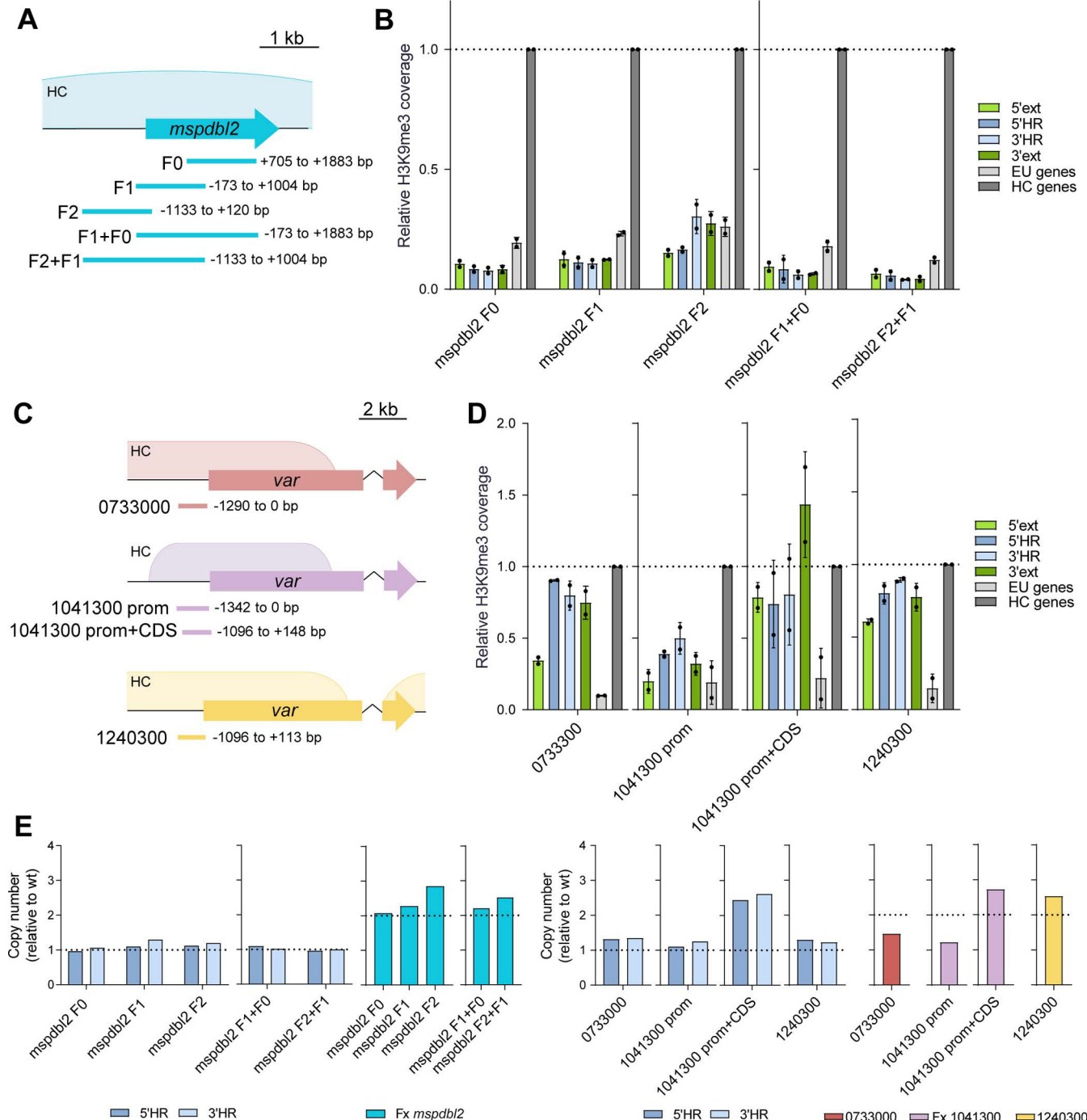

**Fig 5. HC nucleation by sequences from other CVGs.** (A) Schematic (to scale) of the position of *mspdbl2* fragments tested for HC nucleation capacity. The previously described position of the HC domain at this locus is indicated (HC) [16]. (B) H3K9me3 ChIP-qPCR analysis, as in Fig 2C, of transgenic lines carrying *mspdbl2* fragments. Data are presented as the average and s.d. of two biological replicates. (C) Schematic (to scale) of the position of fragments from different *var* genes tested for HC nucleation capacity: 0733000 (from PF3D7_0733000), 1041300 (from PF3D7_1041300) and 1240300 (from PF3D7_1240300). The previously described position of the HC domains at these loci is indicated (HC) [16]. (D) H3K9me3 ChIP-qPCR analysis, as in panel B, of transgenic lines carrying *var* fragments. (E) qPCR analysis of copy number, as in Fig 2D.

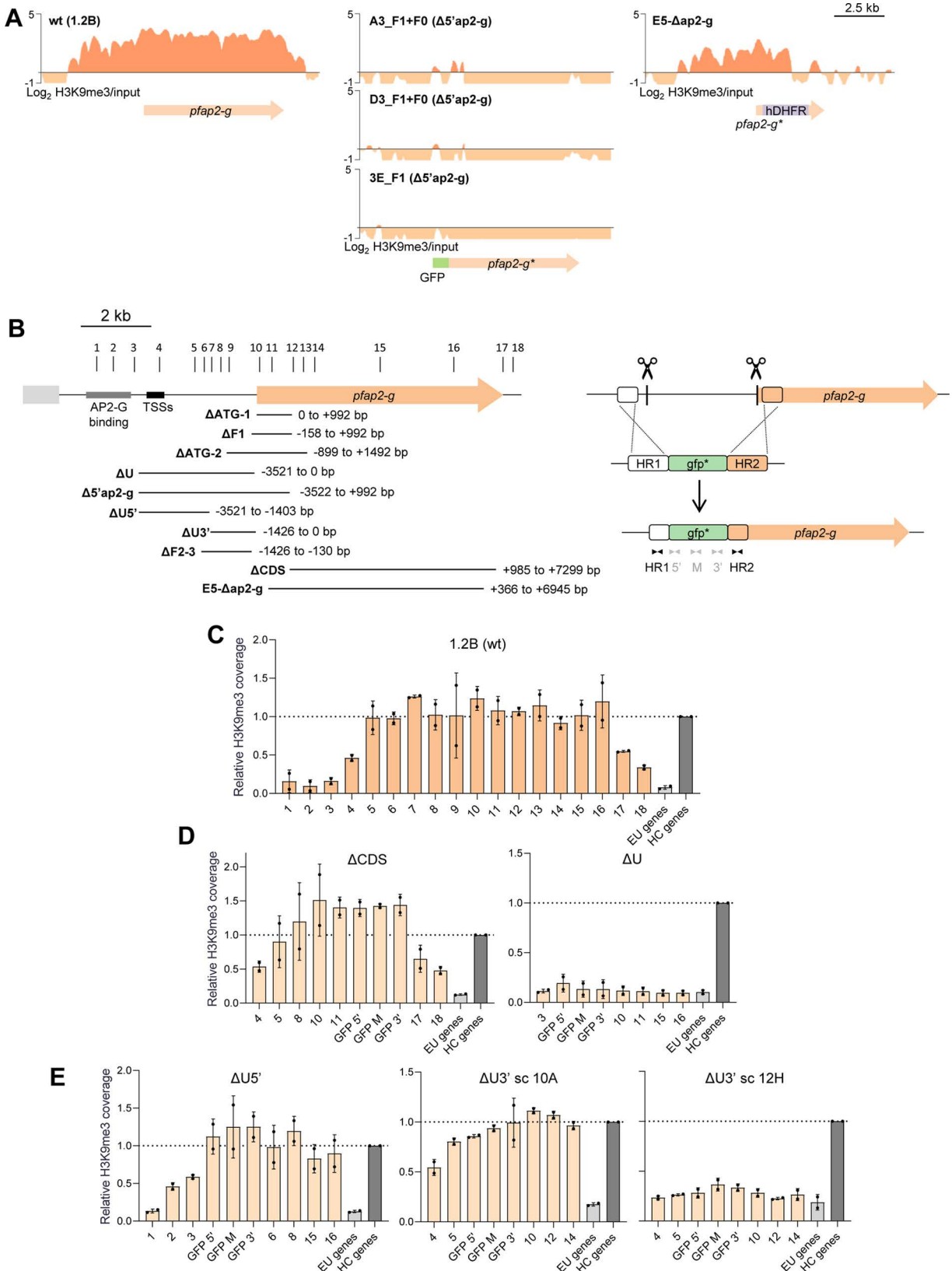

**Fig 6. Analysis of the role of different regions of *pfap2-g* in HC maintenance.** (A) H3K9me3 ChIP-seq profiles at the *pfap2-g* locus in wt (1.2B) and transgenic parasite lines. In experiments with transgenic lines, reads were aligned against the respective edited genome. Block arrows are annotated genes

(start codon to stop codon). (B) Left: Schematic (to scale) of the *pfap2-g* regions deleted to assess their role in HC maintenance. The numbered vertical lines at the top indicate the position of the primer pairs used for ChIP-qPCR analysis. The position of the TSSs and the PfAP2-G binding region is shown. Right: strategy for the deletion of *pfap2-g* regions using CRISPR/Cas9. The scissors indicate the positions targeted by the sgRNAs. A non-functional *gfp* gene (gfp*) was used to replace the deleted region. In this example, part of the upstream region and CDS were deleted, but in other cases, different regions were eliminated. Black and grey arrowheads represent the primers used to assess the number of integrated copies. Primers for gfp* were also used in the ChIP-qPCR analyses. (C) H3K9me3 ChIP-qPCR analysis of the parental wt 1.2B parasite line, from which transgenic lines with deletions at the *pfap2-g* locus were derived. Values are the % input at each position relative to the % input in the positive control heterochromatic genes, as in Fig 2C. Numbers in the x axis are the primer pairs shown in panel B. The dashed line indicates the coverage in the positive control heterochromatic genes. Data are presented as the average and s.d. of two biological replicates. (D) H3K9me3 ChIP-qPCR analysis, as in panel C, of the ΔCDS and ΔU transgenic lines. (E) H3K9me3 ChIP-qPCR analysis, as in panel C, of the ΔU5' transgenic line and the 10A and 12H subclones of the ΔU3' transgenic line.

maintenance at the rest of the locus, deletion of the proximal part (ΔU3') affected HC levels. One ΔU3' subclone had HC at the rest of the locus, whereas another subclone completely lost it (Fig 6E). These results suggest that the proximal part of the *pfap2-g* upstream region plays an important role for HC maintenance, although it is not absolutely essential. We also generated and analyzed several additional transgenic lines, in which different parts of the locus were deleted. The analysis of these additional lines supported the conclusion that the proximal upstream region is important for HC maintenance whereas the other regions are dispensable (Figs 7A and S6). The regions required for HC maintenance showed only a small overlap with those capable of nucleation (Fig 7B). Of note, the majority of these transgenic lines had a single copy of the gfp* gene integrated. In a few cases, plasmid concatemers were integrated in some of the parasites (ΔU5', ΔF1 and ΔF2–3), but analysis of subclones showed that the increased separation between the regions upstream and downstream of the deletion in lines with concatemer integration did not impair HC maintenance (Figs 6, 7 and S6).

## Reestablishment of HC at the *pfap2-g* locus is inefficient

Our experiments based on ectopic fragment integration revealed that *pfap2-g* fragments promote *de novo* HC nucleation inefficiently. This made us predict that if a parasite line had the endogenous *pfap2-g* locus in an euchromatic state, it would take several generations for HC to form again. To test this hypothesis, we used the E5-AP2-G-DD parasite line [39], in which the PfAP2-G protein is fused to a C-terminal FKBP destabilization domain (DD) that, in the absence of the Shld1 ligand, drives protein degradation. The upstream regulatory region of the gene is intact. In this parasite line, when cultured without Shld1, absence of HC at the *pfap2-g* locus would not result in massive sexual conversion and therefore would not pose a fitness cost. Indeed, the E5-AP2-G-DD line shows an unusually high sexual conversion rate when Shld1 is added [39,43], consistent with accumulation of parasites with *pfap2-g* in an euchromatic state during culture without Shld1.

We observed sexual conversion rates of ~45% after a first PfAP2-G stabilization with Shld1 for one cycle, but subsequent stabilization two cycles later resulted in a much lower conversion rate (Fig 8A). After induction with Shld1 for one cycle, culturing in the absence of Shld1 for up to 60 generations led to progressive, partial recovery of high sexual conversion rates when Shld1 was added again (S7 Fig). Together, these results suggest that in cultures always maintained without Shld1, parasites with the *pfap2-g* locus in an euchromatic state progressively accumulated, because this did not pose a fitness cost, and these parasites underwent sexual conversion upon Shld1 addition. The remaining parasites had HC at the *pfap2-g* locus and displayed a regular conversion rate. Consistent with this model, subclones of the E5-AP2-G-DD line (obtained from cultures not exposed to Shld1) had dramatically different sexual conversion rates when Shld1 was added, ranging from <5% to >50% (Fig 8B). As expected, higher sexual conversion was associated with higher *pfap2-g* transcript levels, which increased in the presence of Shld1 as a consequence of the PfAP2-G transcriptional feedback loop [39,43,52,53] (Fig 8C).

ChIP-seq analysis of cultures maintained without Shld1 revealed a lower H3K9me3 coverage at the *pfap2-g* locus in the parental E5-AP2-G-DD line and a subclone with high sexual conversion (B11) compared with a subclone with low conversion (D7) (Fig 8D–F). The H3K9me3 coverage fold-change at the *pfap2-g* locus was among the top differences in the full genome (14th largest D7 vs B11 fold-change, permutation test *p* value = 0.030), with other large differences

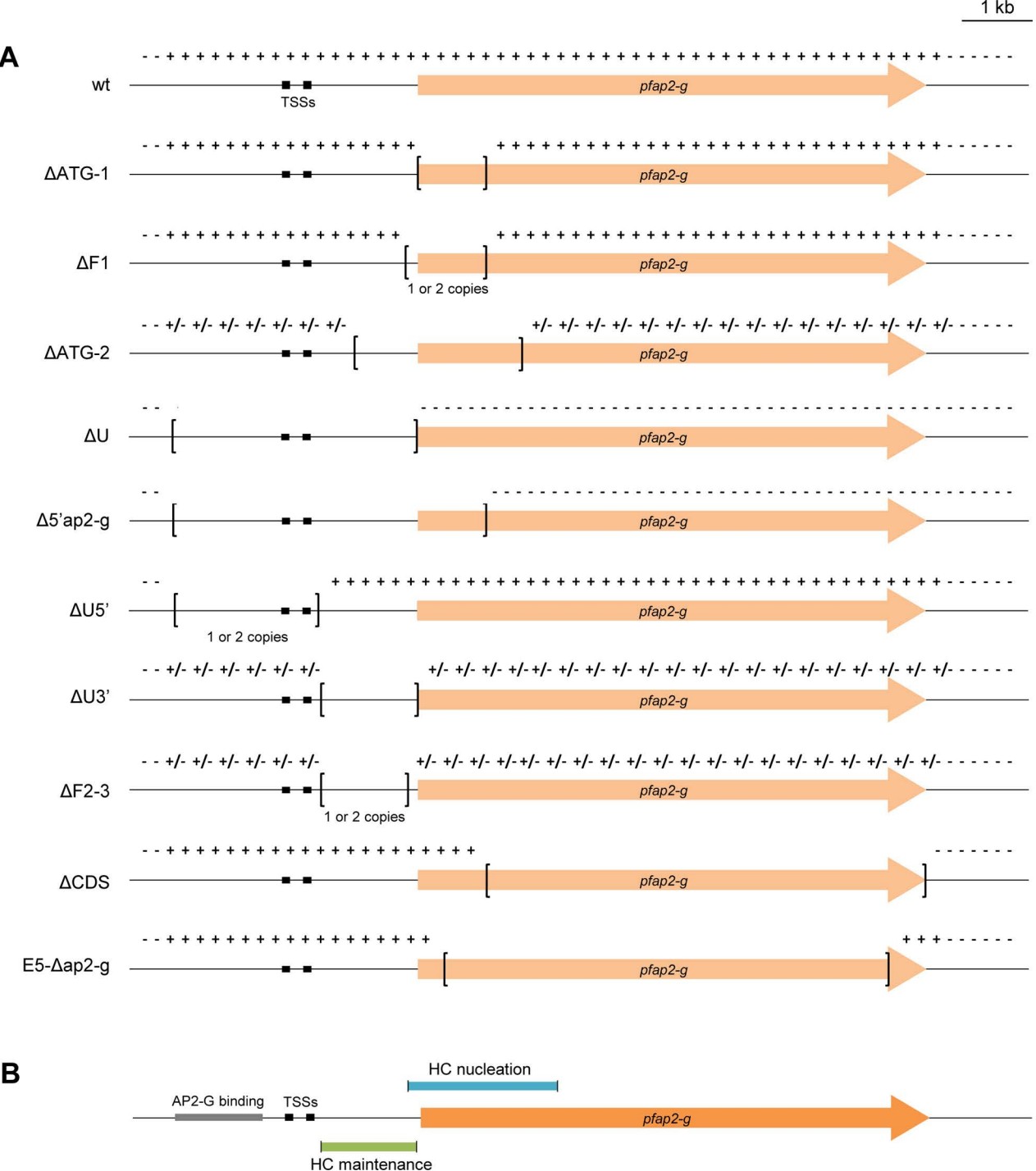

**Fig 7. Summary of HC distribution in parasite lines with different deletions at the *pfap2-g* locus.** (A) Schematic (to scale) of HC distribution, as determined by ChIP-seq or ChIP-qPCR, at the *pfap2-g* locus after deletion of different regions. The deleted region in each transgenic line is indicated by square brackets. TSSs are represented by black boxes. Symbols + and – indicate presence or absence of HC, respectively, whereas +/– indicates partial HC coverage (compared to the wt 1.2B line) in a bulk parasite population or occurrence of HC in some subclones but not others. "1 or 2 copies" refers to the number of gfp* copies integrated. (B) Schematic (to scale) of the relative position of regions of the *pfap2-g* locus capable of nucleating HC or needed for HC maintenance. The position of the TSSs and the PfAP2-G binding sites is shown.

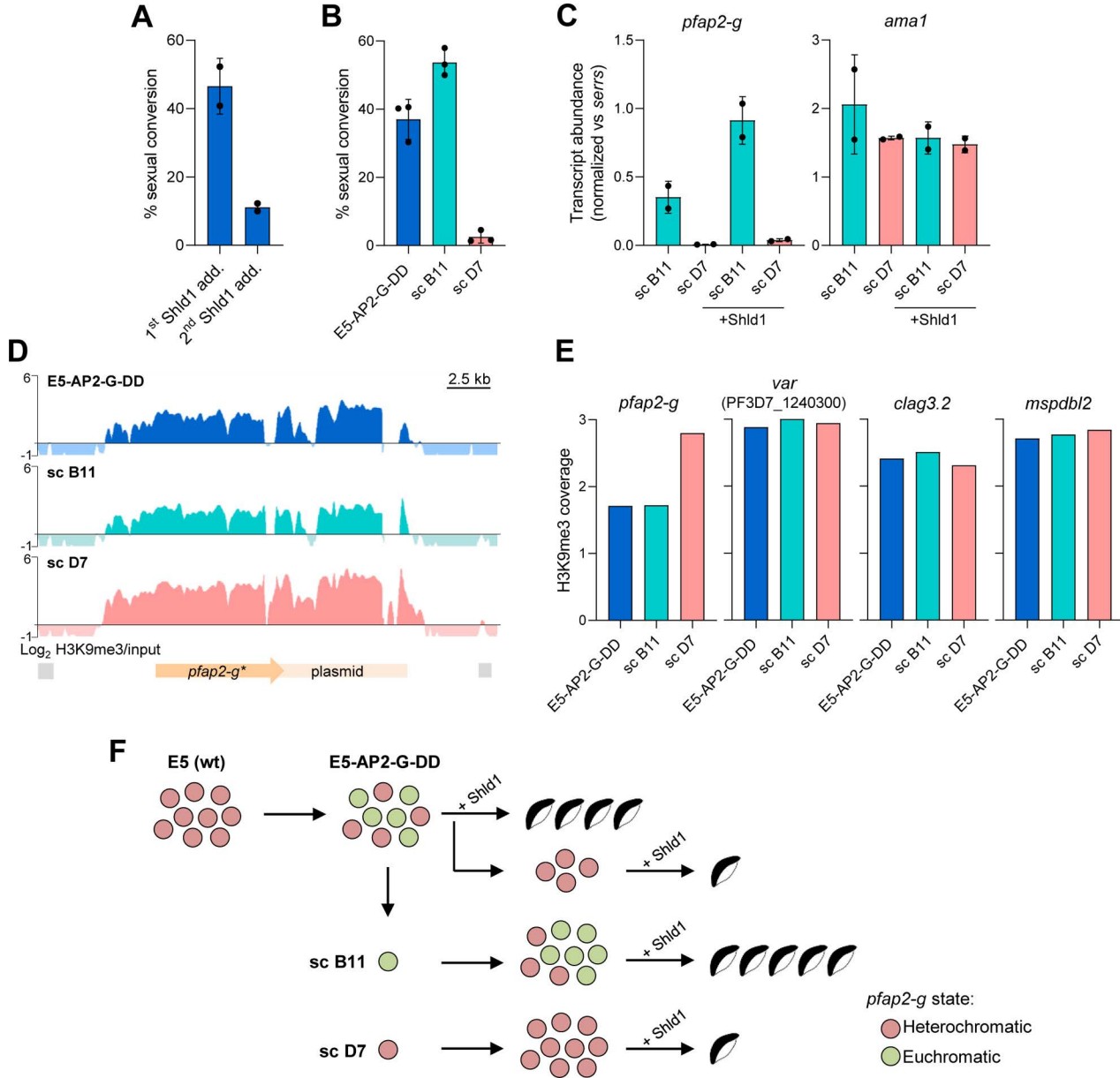

**Fig 8. HC dynamics at the *pfap2-g* locus in the E5-AP2-G-DD line.** (A) Sexual conversion rate of the E5-AP2-G-DD line after exposing cultures to Shld1 for the first time (1st Shld1 add.) and after re-exposing them to Shld1 following 2 growth cycles without Shld1 (2nd Shld1 add.). Data are presented as the average and s.d. of two biological replicates. (B) Sexual conversion rates after Shld1 addition for the parental E5-AP2-G-DD line and its subclones (sc) B11 and D7. Data are presented as the average and s.d. of three biological replicates. (C) Expression of *pfap2-g* and *ama1* (the latter used as a control), determined by RT-qPCR, in E5-AP2-G-DD subclones B11 and D7 at the mature schizont stage (ML10-treated cultures) in the absence or presence of Shld1. Values are normalized by expression of *serrs*. Data are presented as the average and s.d. of two biological replicates. (D) H3K9me3 ChIP-seq profiles at the *pfap2-g* locus in the E5-AP2-G-DD line and subclones B11 and D7. Reads were aligned against the edited genome. Block arrows are annotated genes (start codon to stop codon). (E) Quantification of H3K9me3 ChIP-seq coverage at the *pfap2-g, var* gene PF3D7_1240300, *clag3.2* and *mspdbl2* loci in the parental E5-AP2-G-DD line and its subclones B11 and D7 (no Shld1 added). Average coverage was calculated at the region from -1 kb to +0.5 kb from the start codon of each gene. (F) Model for heterochromatin dynamics at the *pfap2-g* locus and sexual conversion rate (after addition of Shld1) in the E5-AP2-G-DD line and its subclones B11 and D7. Higher number of gametocytes indicates higher sexual conversion rate. Red and green circles represent parasites with heterochromatic or euchromatic *pfap2-g*, respectively. Parasites with euchromatic *pfap2-g* convert into gametocytes upon Shld1 addition, whereas in parasites with heterochromatic *pfap2-g*, the gene is activated and drives sexual conversion in only a small fraction of the parasites at each cycle, as in wild type parasites. After a first treatment of the E5-AP2-G-DD line with Shld1, only parasites with heterochromatic *pfap2-g* continued asexual growth. A second treatment with Shld1 resulted in low levels of sexual conversion.

corresponding mainly to genes involved in antigenic variation (*e.g.*, *var*, *rif*) that are well-known to show HC occupancy variation between subclones [16] (S1 Dataset). This is consistent with the gene being in an euchromatic state in the approximately 50% of parasites that convert upon Shld1 addition in the parental E5-AP2-G-DD line and the B11 subclone. No subclone with 100% conversion upon Shld1 addition was identified, suggesting that when a subclone originated from a single parasite with euchromatic *pfap2-g*, HC formed in some parasites during the time between subcloning and ChIP-seq analysis (24 generations), albeit at low speed consistent with inefficient nucleation. The reduced H3K9me3 coverage in the high conversion subclone was homogeneous across the full *pfap2-g* locus, in contrast with changes upon activation of the gene during natural sexual conversion that involve HC depletion only at the upstream region [30,54]. Together, these experiments suggest that the *pfap2-g* locus can be in an euchromatic state when the protein is non-functional, both the euchromatic and the heterochromatic states are heritable, and spontaneous reestablishment of HC occurs but is inefficient and requires multiple generations (Fig 8F). However, we acknowledge that while this model is consistent with the results of the experiments with the E5-AP2-G-DD line, we do not provide direct evidence for the occurrence of the *pfap2-g* locus in a fully euchromatic state in some parasites or for slow, progressive re-establishment of HC, because we could not analyze subclones at earlier time points.

## Discussion

Transcriptional regulation of key biological processes in *P. falciparum* relies on transitions between HC and euchromatin at loci encoding CVGs [24,25]. However, the molecular determinants for HC formation exclusively in some regions of the genome (*i.e.*, those containing subtelomeric repeats or CVGs) and not others remain largely unexplored. Here, we systematically analyzed how DNA sequence determines the ability of a fragment to nucleate HC *de novo* when ectopically integrated in an euchromatic chromosomal site. We found large differences between fragments of similar size but different sequence in their ability to nucleate HC, which clearly indicates that the primary DNA sequence is a major determinant of where in the genome HC is formed. We also found that, at least for *pfap2-g* sequences, a large total size of nucleation-competent DNA is needed for efficient HC nucleation. Furthermore, maintenance of HC also depends on *cis* sequence elements, as only some specific regions of a HC domain were needed to stably maintain pre-existing HC at the rest of the domain. Of note, the regions needed to maintain pre-existing HC are different from those capable to nucleate HC *de novo* (Fig 7B), consistent with observations in other organisms showing that nucleation and maintenance are separable processes that depend on different factors [1,2]. The PfAP2-G binding sites upstream of *pfap2-g* needed for the PfAP2-G autoregulatory feedback loop [39,52,53] are not located within the regions able to nucleate HC or needed for HC maintenance (Fig 7B). While these regions contain the predicted cognate motifs or experimentally determined binding sites for several ApiAP2 transcriptions factors [55,56], the broad genomic distribution of these factors prevents any conclusion about their participation in HC nucleation or maintenance.

We analyzed the ability to nucleate HC of fragments from an euchromatic gene and from several CVGs located either in large HC blocks (*var* genes) or in small HC islands (*pfap2-g* and *mspdbl2*) [15–18], with a main focus on *pfap2-g*, which encodes the master regulator of sexual conversion [39]. We reasoned that for genes located within large HC blocks, presence of HC may simply reflect HC expansion from neighbor regions or a favorable HC-prone chromosomal environment at the nuclear periphery, whereas in small internal chromatin islands the ability to nucleate HC may be conferred by the primary sequence of the gene. However, all *var* promoter sequences tested nucleated HC efficiently, which indicates that these sequences have an intrinsic capacity to nucleate HC *de novo*, even when placed in an ectopic, previously euchromatic region separated from the subtelomeric or central *var* clusters. In contrast, we found that fragments from the beginning of the *pfap2-g* CDS were only able to nucleate HC inefficiently, and fragments from *mspdbl2* and from the *pfap2-g* upstream region were unable to nucleate. The low efficiency of HC nucleation by *pfap2-g* CDS sequences was reflected in nucleation occurring only in a fraction of the parasites and requiring ~2 kb fragments or multiple copies of ~1 kb fragments, whereas ~1 kb *var* fragments robustly nucleated HC in the vast majority of parasites. These results

argue against nucleation at specific locations of the *pfap2-g* locus being a deterministic event in which the presence of specific sequences always results in rapid formation of HC, and rather suggest that, at least for this locus, nucleation is a probabilistic event. Experiments with the E5-AP2-G-DD line suggest that when the full upstream and coding sequence of the gene is present, HC ends up assembling, but this requires multiple generations and occurs in only a fraction of the parasites. Of note, the contribution of different regions to HC maintenance also appears to be probabilistic, as suggested by differences in HC maintenance between genetically identical subclones. Probabilistic events are a common theme in HC biology, as realized many years ago with the variant expression associated with position effect variegation (PEV) in *Drosophila* [1].

A recent study reported that the *pfap2-g* upstream region was able to form ectopic HC in the context of an artificial chromosome, albeit with an H3K9me3 coverage much lower than in endogenous heterochromatic genes [57]. While these results cannot be directly compared with our study because of the artificial chromosome environment and because different regions were tested, they support our conclusion that *pfap2-g* sequences can nucleate ectopic HC formation, although inefficiently. The ability of *P. falciparum* sequences to recapitulate the epigenetic makeup of their endogenous locus when integrated at an ectopic locus was also reported for H3K48ac [58]. However, this likely operates by a completely different mechanism because, unlike HC, this modification may be a direct consequence of active transcription [23] (associated with the promoter activity of the underlying sequence).

The inefficient *de novo* nucleation of HC by *pfap2-g* fragments and inability to nucleate by *mspdbl2* fragments raises the intriguing possibility that *de novo* nucleation at these (and other) loci may be a very rare event *in vivo*, at least during asexual blood stages. In line with this view, comparative analysis of HC distribution between parasite lines that differ in the expression of many CVGs revealed that the chromatin transitions that result in activation or silencing of these genes typically involve expansion or retraction of pre-existing HC domains, rather than *de novo* formation or complete dismantling [16]. For *pfap2-g*, activation may also involve HC retraction only from the promoter, rather than complete dismantling of the HC domain [30,54]. Furthermore, we show that in a parasite line expressing PfAP2-G fused to a DD domain, in which loss of *pfap2-g* HC does not impair asexual proliferation, re-assembly of HC at this locus was observed but required multiple generations and did not occur in all parasites. This again indicates that HC formation by *pfap2-g* sequences is not deterministic and the HC state of this gene typically depends on inheritance from previous generations rather than on *de novo* formation. However, *de novo* HC formation may occur efficiently at other stages. Epigenetic HC patterns are reset during transmission stages [59–63], and it remains unclear if the reset involves complete dismantling of HC, which would make *de novo* HC nucleation essential, or some parts of each HC domain always persist, dispensing the need for *de novo* nucleation. Altogether, existing data supports a scenario in which HC domains are perpetuated for many generations by inheritance through cell division, and *de novo* nucleation is either not regularly required or required only after transmission.

Although here we did not directly aim to investigate HC spreading or to identify barrier elements that limit spreading, we observed clear evidence for HC spreading in some of the transgenic lines. This affected the gfp* sequence used to replace fragments of the *pfap2-g* locus, which was transfected as naked DNA and after integration became heterochromatic, and the PF3D7_1144400 locus, which was euchromatic but became heterochromatic after integration of fragments competent for HC nucleation. Together with previous reports for *clag3* or *var* genes [64,65], these results indicate that in *P. falciparum* HC can spread into adjacent regions, which is an evolutionarily conserved property of HC [1,2]. Spreading into the PF3D7_1144400 locus had similar limits in different transgenic parasite lines, which is suggestive of the occurrence of barrier elements [11] at these positions. In fact, the limits of naturally-occurring HC domains associated with the active or silenced state of essentially all CVGs are well-defined and conserved in different parasite lines [15,16,20,30], suggesting that barrier elements play a prominent role in the organization of *P. falciparum* HC. How the boundaries of HC domains are established in malaria parasites has not been characterized, but a recent preprint proposed that the 3' end of *pfap2-g* acts as a barrier element [66]. However, the occurrence of barrier elements at the PF3D7_1144400 locus, which in wild type parasites is located far from any HC domain, would be intriguing. An alternative explanation for the absence of

HC spreading beyond the CDS at this locus would be that parasites in which HC spreads into the surrounding intergenic regions may be selected against, because this may affect the expression of the putatively essential neighbor genes.

A limitation of our study is that we could not test the ability to nucleate HC of a larger number of fragments covering the entire loci of the genes studied and additional heterochromatic and control euchromatic genes, because it was not feasible to use our screening approach in a high-throughput manner. This is attributable to the well-known difficulties generating *P. falciparum* transgenic lines, especially when targeting extremely AT-rich non-coding regions, and also to the frequent integration of concatemers, which was found to be a confounding factor for our HC nucleation experiments. Moreover, the low efficiency and non-deterministic nature of HC nucleation by some of the fragments further complicated the analysis, as it implied having to insert very long DNA fragments and to generate and analyze multiple subclones for many of the transgenic lines. The non-feasibility of testing fragments at a larger scale reduced the power to discover potential short DNA motifs with the intrinsic capacity to promote *de novo* HC nucleation, which would have paved the way for the identification of putative DNA binding factors or non-coding RNAs driving nucleation. However, the need of long fragments for efficient nucleation suggests that a single specific short DNA motif able to drive HC nucleation may in fact not exist in *P. falciparum*. Thus, it is possible that multiple different binding sites for putative DNA binding factors or specific DNA physical properties over a long sequence stretch are needed for efficient nucleation. In spite of these limitations, our results demonstrate that the primary sequence of some specific DNA regions contains the information needed to seed HC formation at specific locations of the genome and that different regions are needed to maintain preexisting HC domains.

## Methods

### Parasite cultures and sexual conversion assays

The 3D7 subclones E5 (derived from the 3D7-B stock) and 1.2B (derived from the 3D7-A stock) and the transgenic lines E5-PfAP2-G-DD and E5-Δap2-g have been previously described [39,67]. Parasites were cultured in B+ erythrocytes at 3% hematocrit under standard conditions, with RPMI-1640-based culture medium supplemented with 0.5% Albumax II (Invitrogen). Erythrocytes were obtained from the 'Banc de Sang i Teixits' (Barcelona) after ethical approval by the Hospital Clínic (Barcelona) ethics committee. Routine synchronization was carried out using 5% D-sorbitol lysis to eliminate late asexual stages (trophozoites and schizonts). For the analysis of transcripts by RT-qPCR to validate TSSs, cultures were tightly synchronized by purification of mature schizonts using 63% Percoll (Sigma-Aldrich) gradients followed by sorbitol lysis 5 h later [68].

For the stabilization of PfAP2-G in the E5-AP2-G-DD line (experiments in Fig 8), 0.5 µM AquaShield-1 (Shld1, Cheminpharma) was added to cultures ~10 h after sorbitol synchronization (ring/ early trophozoite stage). To measure sexual conversion rates, at the following cycle after stabilization cultures at the ring stage were treated with 50 mM N-acetylglucosamine (Sigma-Aldrich, A3286) for ≥4 days to eliminate asexual parasites. Light microscopy quantification of Giemsa-stained smears was used to determine sexual conversion rates, calculated as the gametocytemia at day ≥ 4 divided by the initial parasitemia before adding N-acetylglucosamine [43]. For the experiments to measure *pfap2-g* transcript levels (Fig 8C), cultures at the trophozoite stage were treated with ML10 for 22 h before collecting RNA, as previously described [43], to have essentially all parasites at the same mature schizont stage in both subclones.

### 5'RACE and RT-qPCR analysis

TSSs were identified using the FirstChoice 5′ RLM-RACE kit (Thermo Fisher) following the manufacturer's instructions, as previously described [20]. In brief, total RNA from non-synchronized E5 and E5-PfAP2-G-DD (treated with Shld1) cultures was extracted using the TRIzol (Invitrogen) method [68]. Next, RNA was treated with calf intestinal alkaline phosphatase to remove free 5'-phosphate groups from non-mRNA molecules. Then, samples were treated with the tobacco acid pyrophosphatase to remove the 5'-cap structure of full-length mRNAs, allowing the ligation of an RNA adapter with T4 RNA ligase. We used a redesigned RNA adapter (S1 Table) because the one provided with the kit resulted in artefacts that

incorporated exogenous sequences of another chromosome into the *pfap2-g* PCR-amplified product. cDNA synthesis was performed by reverse transcription using random hexamers followed by nested PCR using *pfap2-g* specific primers and primers specific for the RACE adapter (S1 Table). Nested PCR was performed either with the *LA* Taq DNA Polymerase (Takara), with 25 cycles of amplification for each PCR reaction and an annealing temperature of 60ºC, or with the KAPA HiFi PCR Kit (KAPA Biosystems, KK2103), with 20 cycles of amplification in the first PCR and 30 in the second, and an annealing temperature of 62ºC. PCR products were cloned into the pCR-2.1 plasmid using the TA Cloning Kit (Invitrogen) and sequenced with M13 forward and reverse primers.

For RT-qPCR, RNA was extracted using the TRIzol method (Invitrogen), DNAse I-treated (Qiagen), purified using the RNeasy MinElute Cleanup kit (Qiagen) and reverse-transcribed using the AMV Reverse Transcription Kit (Promega) with a mixture of random primers and oligo (dT). Transcript abundance was measured by real-time quantitative PCR (qPCR) in triplicate wells using the standard curve method, with the Power SYBR Green Master Mix (Applied Biosystems) and a 7900HT Fast Real-Time PCR System, as previously described [68]. Using the standard curve method, results are not affected by primer efficiency. However, all primer pairs had an efficiency >80%. Transcript levels of the serine-tRNA ligase (*serrs*, PF3D7_0717700) were used for normalization. All primers used for RT-qPCR analysis are described in S1 Table.

## Plasmids

The pL7-KO-ap2g plasmid (S8A Fig) used to delete the *pfap2-g* upstream region and part of the CDS (Δ5'ap2-g line) was derived from the pL6-egfp-yfcu plasmid [69]. The original HR2 was removed by digestion with *EcoR*I and *Nco*I, followed by treatment with T4 DNA polymerase (NEB) to generate blunt ends and religation with T4 DNA ligase (Roche). The *pfap2-g* HR1 (positions -3929 to -3521 bp relative to the start codon) was PCR amplified and cloned into *Not*I/*Pst*I sites, disrupting the *yfcu* cassette. The *pfap2-g* HR2 (positions +991 to +1973 bp) was amplified and cloned into *Spe*I/*Afl*II sites, replacing the original HR1. To add the sgRNA, two annealed oligonucleotides containing the guide sequence (recognizing a sequence located at the *pfap2-g* -3503 to -3522 bp position) were cloned into a *Btg*ZI site using the In-Fusion HD Cloning Kit (Clontech). Last, an incomplete *gfp* sequence (gfp*), lacking the start codon, was amplified and cloned between HR1 and HR2, removing the remaining part of the *yfcu* cassette, using *Pst*I/*Spe*I sites. To generate plasmid pDC2-Cas9-hDHFR-ap2g-3' (S8A Fig) expressing a second sgRNA in addition to Cas9 and the *hdhfr* selectable marker, a guide RNA covering positions +921 to +940 bp was cloned into a *Bbs*I site of plasmid pDC2-Cas9-hDHFR [70] using the InFusion system.

For the integration of the fragments into the PF3D7_1144400 locus, pL1144400-Fx plasmids ("x" indicates each different fragment; S4A Fig) were generated from the pL6-egfp-yfcu plasmid [69]. The original HR2 was removed as described above. The PF3D7_1144400 HR1 (positions +1144 to +1562 bp from the start codon) was amplified with primers incorporating *Nco*I + *Spe*I sites in the reverse primer and a *Sac*II site in the forward primer. The PF3D7_1144400 HR2 (positions +1646 to +2199 bp) was amplified incorporating *Spe*I and *Afl*II sites, and cloned together with HR1 into *Sac*II/*Afl*II sites, replacing the original HR1. The original *hdfhr* and sgRNA expression cassettes were removed using restriction sites *Aat*II and *Afl*II, blunt ending and religation. The *yfcu* cassette was removed using restriction sites *Not*I and *Sac*II, blunt ending and religation. However, for all fragments except ~1Kb fragments in Fig 2 and shorter fragments (S3 Fig), the selection cassette was reintroduced using the same restriction sites (pL1144400-Fx-yFCU plasmids) (S9 Fig). Fragments (Fx) from different genes were PCR amplified and cloned into *Nco*I/*Spe*I sites with the primers listed in S1 Table. To generate the plasmid pDC2-Cas9-hDHFRyFCU-1144400 (S4A and S9 Figs), used for transfections with pL1144400-Fx and pL1144400-Fx-yFCU plasmids, a guide RNA (covering positions +1566 to +1585 bp from the PF3D7_1144400 start codon) was cloned into a *Bbs*I site of plasmid pDC2-Cas9-hDHFRyFCU [71] using the InFusion system.

To generate the plasmid pL-ap2g_Del1_GFP_yfcu, used to obtain the ΔATG-1 line, the original HR1, *ama1* fragment and HR2 regions were removed from the pL11444000-ama1-yFCU plasmid (S9 Fig) by digestion with *Sac*II and *EcoR*I. The HR1 and HR2 were cloned by InFusion, introducing *Hind*III and *Spe*I restriction sites between them to clone the gfp*

sequence by ligation (S10A Fig). For the ΔATG-2, ΔCDS, ΔU, ΔU5', ΔU3' and ΔF2–3'-A lines, HRs for each construct were amplified, introducing *Afl*II and *Hind*III restriction sites between HR1 and HR2, and cloned into *Sac*II/*Nco*I sites of a plasmid derived from pL6-eGFP-yFCU [69] using the InFusion system. Digestion of the plasmid with *Sac*II and *Nco*I removed the original HRs and the *hdhfr* sequence. Next, a different sgRNA for each transgenic line was cloned into a *Btg*ZI site. Finally, the gfp*sequence was cloned into the *Afl*II/*Hind*III sites between HR1 and HR2 by ligation, resulting in pL-7-ap2g_Delx_GFP_yfcu plasmids (where "x" refers to each fragment deleted) (S10A Fig). Finally, a second sgRNA used for each transgenic line was cloned into a *Bbs*I site of plasmid pDC2-Cas9-hDHFR [70], resulting in pDC2-Cas9-hDHFR-sgRNA-Delx plasmids (S10A Fig and S1 Table).

For deletion ΔF1 and to generate a second transgenic line with the deletion ΔF2–3' (ΔF2–3'-B line), we followed an alternative three-plasmid strategy (S10B Fig). Donor plasmids pUC19-KO-Fx were generated by simultaneously cloning the respective HR1 and HR2 for each construct and the gfp* sequence into a *Bam*HI site of the pUC19 vector, using the InFusion system. Guides were cloned into the pDC2-Cas9-hDHFR plasmid as described above.

All guides were designed and selected using the EuPaGDT web-based tool [72]. PCR amplification from *P. falciparum* gDNA was performed using LA Taq DNA Polymerase (Takara). For plasmid cloning and amplification, we used *Escherichia coli* DH5α or Stellar Competent Cells (Clontech) for difficult cloning. Oligonucleotides were purchased from Integrated DNA Technologies (IDT). All primers and oligonucleotides are described in S1 Table.

## Generation of transgenic lines

Unless otherwise stated, transfections were performed according to standard procedures [73], electroporating cultures at the ring stage with a BioRad GenePulser Xcell electroporator. Starting 15h after transfection, cultures were selected for 4 days with 10 nM WR99210 (Jacobus Pharmaceuticals). When parasites were observed, for transfections with donor plasmids containing the *yfcu* negative selection marker, cultures where selected with 1 μM 5-fluorocytosine (5-FC; clinical grade Ancotil, Mylan N.V.) for 8 days, to remove parasites carrying episomal copies of the plasmids or integration of concatemers containing the full plasmid. Note that in the case of the *var* 1240300 line, cultures were analyzed without 5-FC selection (for comparison with transgenic lines generated with plasmids lacking the *yfcu* selection marker, Fig 2), and after 5-FC selection (Fig 5). Diagnostic PCR to assess correct edition was performed using the *LA* Taq DNA Polymerase (Takara), typically with primers external to the region edited. Diagnostic qPCR using the standard curve method was performed to assess the presence of plasmid regions not intended to be integrated and to determine the relative copy number of HRs and integrated fragments. Analysis by qPCR was performed in triplicate wells as described above for RT-qPCR. We used gDNA from the W4-2 (containing a single copy of the *hdhfr* gene and plasmid backbone) or the wt 3D7-A parasite lines [74] for the standard curve included in each plate for each primer pair. In all cases, copy number was determined relative to a single copy gene (*serrs*). The primers used for diagnostic PCR and qPCR are described in S1 Table. Additionally, for the parasite lines containing integrated fragments F1 or F1+F0 analyzed by ChIP-seq, we used the input reads to characterize their genome. Alignment against the edited genomes confirmed correct integration at the PF3D7_1144400 locus, with multiple PF3D7_1144400/*pfap2-g* hybrid reads spanning the integration junctions. These parasite lines were subcloned after transfection, and hence are genetically homogeneous, excluding the possibility that the fragments integrated elsewhere in a subset of the parasites.

To obtain the Δ5'ap2-g line, 1.2B cultures were transfected with 60 μg of plasmid pDC2-Cas9-HDHFR-ap2g-3' and 12 μg of pL7-KO-ap2g linearized using a *Sca*I site located in the backbone of the plasmid. Subclones were obtained by limiting dilution and assessed by diagnostic PCR (with primers external from the edited region) and qPCR analysis to assess presence of plasmid sequences not intended to be integrated (*hdhfr* marker). The H11 subclone was selected for further work because it contained no episomal copy of the transfection plasmids or multiple integrations (S8B and S8C Fig).

For the integration of fragments at the PF3D7_1144400 locus, Δ5'ap2-g (H11 subclone) cultures were transfected with 60 μg of pDC2-Cas9-yFCU-hDHFR-1144400 and 12 μg of either pL1144400-Fx (S4 Fig) or pL1144400-Fx-yFCU (S9 Fig), linearized using the *Sca*I or the *Xmn*I restriction sites.

To generate transgenic lines with deletions at the *pfap2-g* locus, 1.2B cultures were transfected with 60 µg of pDC2-Cas9-yFCU-hDHFR-sgRNA-Delx and 12 µg of either pL-ap2g_Del1_GFP_yfcu (ΔATG-1, the only transgenic line that was generated using only one guide RNA) or pL-7-ap2g_Delx_GFP_yfcu (ΔATG-2, ΔCDS, ΔU and ΔU5') plasmids, linearized with *Xmn*I, *Sca*I or *Pvu*I (S10A Fig). The exception were the lines carrying the ΔF2–3'-B and ΔF1 deletions, which were generated using a three-plasmids transfection with 12 µg of pUC19-KO-Fx (linearized using *Xmn*I) and 60 µg of each of two different pDC2-Cas9-hDHFR-sgRNA-Delx plasmids containing different guides (S10B Fig). This was followed by subcloning by limiting dilution, when required, to obtain pure populations of edited parasites. In two cases in which we could not obtain edited parasites using regular transfections (ΔU3' and ΔF2–3-A, the latter is equivalent to ΔF2–3-B, with the same F2-3 region deleted, but generated differently), we used Nucleofactor transfection of schizonts. For this, cultures were tightly synchronized to a 2 h age window using Percoll purification followed by sorbitol lysis 2 hours later. When most parasites reached the mature schizont stage, a new Percoll purification was performed, followed by transfection using the AMAXA P3 primary cell 4D Nucleofector X Kit L (Lonza), according to manufacturer's instructions, with 60 µg of pDC2-Cas9-hDHFR-gRNA-Delx plasmid and 60 µg of pL-7-ap2g_Delx_GFP_yfcu, linearized with *Sca*I or *Pvu*I.

## Southern blot

Southern blot was performed according to standard procedures [74]. gDNA was digested with the restriction enzymes *Aat*II and *Hinc*II, plus *Afl*II to cleave episomal plasmids. Digestions were resolved in 1.25% agarose gels and transferred to Amersham Hybond N+ nylon membranes (GE Healthcare) for detection with a specific probe labelled with $\alpha$-$^{32}$P dATP (Perkin Elmer). The primers used to amplify the probe were the same as the ones used for the amplification of the HR1 of PF3D7_1144400 (S1 Table). Hybridization and washes were performed at 62°C.

## ChIP-qPCR and ChIP-seq

Chromatin extraction from cultures at the late trophozoite/schizont stage was performed as previously described, using the MAGnify Chromatin Immunoprecipitation System (Life Technologies) [30,47,64], with minor modifications. Briefly, for ChIP-qPCR, formaldehyde-crosslinked samples were sonicated using a Bioruptor Plus sonication device (Diagenode) at high power, with 6 cycles of 30 seconds ON/ 30 seconds OFF followed by centrifugation and recovery of the aqueous phase, which was sonicated for 2 additional cycles of 30 seconds ON/ 30 seconds OFF to obtain sheared chromatin with a size around 400–500 bp. Immunoprecipitation was performed overnight at 4ºC with 0.5 µg of chromatin and 1 µg of antibodies against H3K9me3 (Diagenode), previously coupled to protein A/G magnetic beads provided in the MAGnify kit. Washing, de-crosslinking and elution were performed following the kit recommendations. Eluted DNA was diluted 1:2 for further downstream analysis by qPCR. For qPCR analysis, samples were analyzed in triplicate wells using the standard curve method as described above for RT-qPCR, with a standard curve in each plate for each primer pair (primers described in S1 Table). The standard curve method [68] prevents inaccuracies related with low primer efficiency, but all primer pairs had an efficiency >80%.

For ChIP-seq [47], samples were sonicated using an M220 sonicator (Covaris) at 10% duty factor, 200 cycles per burst, 140 W of peak incident power for 10 min. Immunoprecipitation was performed overnight at 4ºC with 2 µg of chromatin and 4 µg of antibodies against H3K9me3 (Diagenode) or H3K9ac (Diagenode), previously coupled to protein A/G magnetic beads provided in the MAGnify kit. Washing, de-crosslinking and elution were performed following the kit recommendations, but avoiding high temperatures that may result in denaturation of extremely AT-rich intergenic regions: de-crosslinking, proteinase K treatment and elution were performed at 45ºC (for 2 h, overnight, and 1.5 h, respectively).

Libraries for Illumina sequencing were prepared from 5 ng of immunoprecipitated DNA using a protocol adapted for extremely AT-rich genomes [45,47]. In brief, after end-repair and addition of 3′ A-overhangs, NEBNext Multiplex Oligos for Illumina (NEB, E7500) were ligated. Purification steps were performed with Agencourt AMPure XP beads (Beckman Coulter, A63880). Libraries were amplified using the KAPA HiFi PCR Kit (KAPA Biosystems, KK2103) in KAPA HiFi Fidelity

Buffer with the following conditions: 95°C for 3 min; 9 cycles at 98°C for 20 s and 62°C for 2 min 30 s; and 62 °C for 5 min. Amplified libraries were purified using 0.9X AMPure XP beads to remove adapter dimers. The library size was analysed in a 4200 TapeStation System (Agilent Technologies). We obtained 12–16 million 125 bp paired-end reads per sample using the HiSeq2500 or the NextSeq 500 System (Illumina).

The analysis of ChIP-seq data was performed as previously described [16,30]. After initial processing, the reads where aligned to the reference 3D7 genome modified when required to include the alterations introduced after genome editing (i.e., deletion of the *pfap2-g* upstream region and part of the CDS, deletion of the CDS, integration of fragments at the PF3D7_1144400 locus or integration of a plasmid at the *pfap2-g* locus to add a C-terminal DD tag). After full processing, data was visualized using IGV (v2.4.10) [75]. To calculate genome-wide HC occupancy at genes, H3K9me3 coverage was quantified at the region -1000 to + 500 bp from the start codon of each gene, as previously reported [16,30]. To calculate the statistical significance of the changes in HC coverage at the *pfap2-g* locus, we used a permutation test. For this, we generated a null distribution for the coverage fold-change between D7 and B11 in all clonally variant genes [61] by generating 10,000 permutations of the fold-change values and compared the observed ranking of each gene with the ranking in the null distribution.

## Supporting information

**S1 Fig. RT-qPCR validation of the *pfap2-g* TSSs.**
(PDF)

**S2 Fig. Parasite lines used in this study and H3K9me3 ChIP-qPCR recovery (% input) values.**
(PDF)

**S3 Fig. Assessment of HC nucleation by small *pfap2-g* fragments derived from fragments F0 and F1.**
(PDF)

**S4 Fig. Integration of ~1 kb fragments into the PF3D7_1144400 locus.**
(PDF)

**S5 Fig. Assessment of HC nucleation in subclones of parasite lines carrying *var* fragments.**
(PDF)

**S6 Fig. Analysis of the role of different regions of *pfap2-g* in HC maintenance.**
(PDF)

**S7 Fig. Evolution of the sexual conversion rate of E5-AP2-G-DD after a second exposure to Shld1 at different times.**
(PDF)

**S8 Fig. Generation of the Δ5'ap2-g line.**
(PDF)

**S9 Fig. Generation of transgenic lines to assess HC nucleation by single copies of integrated fragments.**
(PDF)

**S10 Fig. Generation of the transgenic lines to study HC maintenance at the *pfap2-g* locus.**
(PDF)

**S1 Table. Oligonucleotides used in this study.**
(PDF)

**S1 Dataset. Genome-wide heterochromatin coverage in subclones of the E5-AP2-G-DD line with different sexual conversion rates.**
(XLSX)

## Acknowledgments

We thank L. Michel-Todó for his contribution to the bioinformatic analysis, C. Bancells for providing cDNA samples of E5 and E5-PfAP2-G-DD, N. Rovira-Graells for technical assistance with Southern blot analysis, J. J. López-Rubio (University of Montpellier) for plasmid pL6-egfp-yfcu, M. Lee (Wellcome Sanger Institute) for plasmid pDC2-Cas9-HDHFR and E. Knuepfer (The Francis Crick Institute) for plasmid pDC2-Cas9-hDHFRyFCU.

## Author contributions

**Conceptualization:** Alba Pérez-Cantero, Oriol Llorà-Batlle, Alfred Cortés.

**Formal analysis:** Alba Pérez-Cantero, Oriol Llorà-Batlle, César Martínez-Guardiola.

**Funding acquisition:** Alfred Cortés.

**Investigation:** Alba Pérez-Cantero, Oriol Llorà-Batlle, Ingrid Pelaez-Conde.

**Methodology:** Alba Pérez-Cantero, Oriol Llorà-Batlle, César Martínez-Guardiola.

**Supervision:** Alfred Cortés.

**Visualization:** Alba Pérez-Cantero, Oriol Llorà-Batlle.

**Writing – original draft:** Alba Pérez-Cantero, Oriol Llorà-Batlle, Alfred Cortés.

**Writing – review & editing:** Alba Pérez-Cantero, Oriol Llorà-Batlle, Ingrid Pelaez-Conde, César Martínez-Guardiola, Alfred Cortés.

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
