## [Decision Letter · Decision Letter 0]

26 Feb 2025

PPATHOGENS-D-25-00221

Heterochromatin de novo formation and maintenance in Plasmodium falciparum

PLOS Pathogens

Dear Dr. Cortes,

Thank you for submitting your manuscript to PLOS Pathogens. After careful consideration, we feel that it has merit but does not fully meet PLOS Pathogens's publication criteria as it currently stands. Therefore, we invite you to submit a revised version of the manuscript that addresses the points raised during the review process.

You will see that all three reviewers were very positive about the study and are overall enthusiastic about its ultimate publication. However, each had suggestions about various aspects of data interpretation and presentation. I imagine that the reviewers' suggestions can be addressed with some additional analysis and discussion and with little additional experimentation. We look forward to seeing your revision. 

Please submit your revised manuscript within 30 days Apr 27 2025 11:59PM. If you will need more time than this to complete your revisions, please reply to this message or contact the journal office at plospathogens@plos.org. Please include the following items when submitting your revised manuscript:

We look forward to receiving your revised manuscript.

Kind regards,

Kirk W. Deitsch

Academic Editor

PLOS Pathogens

Margaret Phillips

Section Editor

PLOS Pathogens

Sumita Bhaduri-McIntosh

Editor-in-Chief

PLOS Pathogens

orcid.org/0000-0003-2946-9497

Michael Malim

Editor-in-Chief

PLOS Pathogens

orcid.org/0000-0002-7699-2064

**Journal Requirements:**

https://journals.plos.org/plospathogens/s/submission-guidelines#loc-parts-of-a-submission

- ® on page: 24

- TM on page: 24.

5) We have noticed that you have uploaded Supporting Information files, but you have not included a list of legends. Please add a full list of legends for your Supporting Information files after the references list.

Potential Copyright Issues:

i) Figures 2A, 6B, and S4C. Please confirm whether you drew the images / clip-art within the figure panels by hand. If you did not draw the images, please provide (a) a link to the source of the images or icons and their license / terms of use; or (b) written permission from the copyright holder to publish the images or icons under our CC BY 4.0 license. Alternatively, you may replace the images with open source alternatives. See these open source resources you may use to replace images / clip-art:

7) Please amend your detailed Financial Disclosure statement. This is published with the article. It must therefore be completed in full sentences and contain the exact wording you wish to be published.

1) State what role the funders took in the study. If the funders had no role in your study, please state: "The funders had no role in study design, data collection and analysis, decision to publish, or preparation of the manuscript.".

**Reviewers' Comments:**

Reviewer's Responses to Questions

**Part I - Summary**

Reviewer #1: In this manuscript, the authors investigate DNA sequence determinants of heterochromatin nucleation/maintenance in P. falciparum, which has been largely unexplored. They provide novel and well-controlled evidence that specific sequences from var genes and ap2-g are able to nucleate heterochromatin at a normally euchromatic locus. They go on to elucidate which sequences are needed at the endogenous ap2-g locus to maintain heterochromatin. It is interesting how the data support different roles – HC nucleation versus maintenance – for different DNA elements. I appreciate the difficulty of all these experiments, especially the genome editing in AT-rich regions, incorporation of multiple copies, etc. The authors were very careful in their experiments, always aware of the limitations of their system, and never overstate their findings. The authors readily admit the main weakness I found with this study, being that the fragments investigated were too long to provide insight into which DNA-binding factors or ncRNAs might facilitate their described heterochromatin nucleation/maintenance phenotypes. However, the data are convincing/conclusive, interesting, and provide solid evidence for the role of these DNA elements in heterochromatin formation in the most “real” way possible (i.e. direct modification of endogenous loci, without the use of artificial chromosomes, etc). I feel that their findings would be useful to the community studying epigenetics at large, and particularly for those interested in heterochromatin regulation in P. falciparum.

Reviewer #2: The authors have undertaken a thorough study investigating a difficult question that has long puzzled malaria researchers. Their demonstration of sequence dependent nucleation and maintenance of heterochromatin is important and useful.

Reviewer #3: In this study, the authors aimed to understand how heterochromatin is nucleated and maintained in the human malaria parasite P. falciparum. Heterochromatin in malaria parasites is critical for controlling key pathogenicity mechanisms such as antigenic variation and several erythrocyte invasion ligands. Another key mechanism directed by heterochromatin is the developmental decision between asexual and sexual parasite fate, which is controlled by the transcription factor AP2-G that is heterochromatin packaged in asexual parasites, but upon destabilization of heterochromatin at its promoter becomes activated and triggers the sexual differentiation program. To understand the sequence requirements for heterochromatin nucleation with a focus on AP2-G, the authors integrated various sequence elements form the AP2-G coding and promoter region (also some other control genes) into a non-essential euchromatic gene locus and assayed the heterochromatin state by chromatin immunoprecipitation. In addition, to gain insight into heterochromatin maintenance, they also modified the endogenous AP2-G locus to define sequences required for maintenance of the pre-established heterochromatin status. This led to the narrowing down of two relatively broad and overlapping regions that are supporting nucleation and maintenance, respectively. However, the authors also demonstrate that there is significant clonal variability, indicating that these sequences are not strictly driving these processes.

**Part II – Major Issues: Key Experiments Required for Acceptance**

Reviewer #1: I appreciate that the authors discuss probabilistic heterochromatinization of loci such as ap2-g. The authors see quite a bit of variability between clones in this study. In Fig. 6E, two different clones of delta U3’ show very different phenotypes. This is true for the sexual conversion experiments as well. This suggests that H3K9me3 levels are rather dynamic, and that maybe at certain loci, it’s not simply a question of euchromatin versus heterochromatin, but dynamism of H3K9me3 levels…more like a constant balance between the writers and erasers. However, I think the authors down-play their findings a bit with regard to heterochromatin nucleation driven by ap2-g upstream sequences. Yes, the var gene sequences are more efficient at nucleating heterochromatin, but I would not draw the conclusion that “nucleation at the pfap2-g locus [is not] a deterministic event in which the presence of specific sequences always results in formation of HC”. In the AP2-G-DD background, I’m assuming that the entire upstream region of ap2-g is present, including the sequence that the authors determined was sufficient to drive heterochromatin nucleation at the 1144400 gene. Although lower than the H3K9me3 levels at ap2-g in D7, B11 still has plenty of H3K9me3 there and does not show 100% conversion. Thus, I would not say that this locus is euchromatinized in B11. The authors claim in the discussion that heterochromatin re-establishment at ap2-g in this line takes multiple generations, but they do not provide H3K9me3 ChIP time course data to support this. The key experiment to support the claims that ap2-g cis sequences are not deterministic would be to delete the upstream HC-nucleating sequence of ap2-g in clone B11 to see if this would truly deplete H3K9me3 levels at the locus and lead to 100% conversion.

Reviewer #2: The study was somewhat affected by the difficulty of cloning the low complexity intergenic sequences and the uncertainty of their recombination. Overall I think the multiple experiments conducted control for these variables and extra experimentation to address them is not required although some analysis of their existing sequencing data might address concerns about off target integration or the presence of episomes/concatamers.

Line 242

“HC formation at the F0 and F1 fragments integrated in the genome is demonstrated by the detection of H3K9me3 with the external primer pairs 5’ext. and 3’ext., which recognize sequences adjacent to the site of integration that are not present in the transfection plasmids (Fig. 2A,C).”

This is true albeit the ext sequences are less enriched with H3K9me3 than the cloned sequence. How can the authors exclude the possibility of off target integration? It looks like F0 and F1 might be contained within the endogenous recombinant ap2g locus. Could there have been a low frequency of recombination here? The intergenic sequences could have recombined with other low complexity high %AT sequence in the genome but this doesn’t matter because they failed to nucleate heterochromatin. This could be very difficult to resolve but a simple thing to try would be mapping the F1 input from fig4C (which has multiple integrated copies in fig 2D) to the various possible in silico modified recombinant loci. The authors could also filter for the inserted sequences and de novo assemble then map the assembled sequences flanking the cloned ap2g sequences back onto the genome to see whether the sequences might have integrated elsewhere.

Reviewer #3: - The experiments integrating sequences of the AP2-G locus into the euchromatic Pf3D7_11444000 gene nicely delineate a sequence at the gene start to be capable of nucleating heterochromatin. However, the sequence is less efficient than parts of the 5’ region of var genes. I was wondering: could the inefficient heterochromatin detection not also be a result of the absence of sequences from area -899 to -130, which were identified as critical for heterochromatin maintenance in Fig. 6 and 7? If I understand correctly, these sequences are missing in the clones carrying F1 or F1+F0.

- Sublcones were generated for U3’, presumably because the original line still contained wt DNA according to Figure S10. As the clonal data for both, nucleation (F1+F0 or F1) and maintenance (U3’) demonstrate a substantial degree of clonal variability, how confident are you that you don’t miss an effect for the other (uncloned) regions?

- AP2-G is known to bind to its own promoter and enhance its own transcription. Are there particular sequence motifs evident in the areas for nucleation and maintenance (-899 to -130)? Where are the multiple AP2-G binding sites relative to the identified regions? Are any other predicted TF binding sites or motifs evident? In line 260, the authors mention sequence repetitiveness as a potential factor for making nucleation more effective. Therefore, I think analysis of the primary sequence may help in understanding the results better.

**Part III – Minor Issues: Editorial and Data Presentation Modifications**

Reviewer #1: • Fig S2: It is unclear what the difference is between the left and right graphs. Are they for different clones or replicates?

• Fig. 2C: I always prefer to see raw values rather than ratios. Maybe the authors could put bar graphs in supplementary data with raw values for each primer pair used, including separate bars for each heterochromatic and euchromatic control locus? This is just to see the variability amongst the different loci for one experiment, not necessary for the rest of the manuscript. Since they only have two replicates, I think it would be more transparent to provide each data point on the bar graph rather than have error bars.

• Could the authors please provide a figure (or just add to Fig. 2B) showing H3K9me3 and/or HP1 ChIP-seq data (already published is fine) enrichment over the ap2-g and 1240300 loci? Just to give a visual of the normal chromatin composition of the fragments tested.

• Have the authors tested the primer efficiencies used for ChIP-qPCR to confirm that the different levels of enrichment aren’t due to differences in AT-richness. For example, regions up- and downstream of the HR/internal Fx sequences might be less GC rich and lead to less relative amplification.

• Fig. 4E/F: It’s interesting that H3K9me3 coverage tapers off just at the start and end codons of the 1144400 gene. The authors discuss how barrier elements are well defined for normal heterochromatin domains, but could the authors comment on why they think HC doesn’t spread into the upstream/downstream regions of this euchromatic gene?

• Could the authors please provide western blots of AP2-G-DD for the different subclones in Fig. 8 (with and without shield) to show that the differences in H3K9me3 levels at the ap2-g locus correspond to different levels of AP2-G expression?

Reviewer #2: Line 259

“Formation of HC only in parasite lines with multiple copies of F0 or F1 may be explained by the larger total size of putative HC-promoting DNA or by sequence repetitiveness (occurrence of the same sequence more than one time), as repetitiveness itself may promote HC formation. In Drosophila and other organisms, presence of multiple copies of a transgene is sufficient to trigger HC-based silencing [48, 49]. To distinguish between these possibilities”

Another possibility is off target integration into a heterochromatic locus. I couldnt see a plasmid map of the pDC2 plasmid and the earliest citation I could find DOI: 10.1038/NMICROBIOL.2016.166 didn’t have a plasmid map. Does the plasmid contain other P falciparum sequences that could mistarget the integration?

Line 288

“These results confirm that the intermediate levels of H3K9me3 observed in transgenic lines with HC-positive pfap2-g fragments reflect population heterogeneity rather than partial H3K9me3 occupancy in individual parasites”

I don’t think the data excludes the possibility that clones can have differing levels of H3K9me3 at a locus, the two heterochromatic ap2g-f1 subclones have differing levels of heterochromatin and the individual clones have different levels at the different pcr sites.

Line 388

“ChIP-seq analysis of cultures maintained without Shld1 revealed a lower H3K9me3 coverage at the pfap2-g locus in the parental E5-AP2-G-DD line and a subclone with high sexual conversion, compared with a subclone with low conversion. This is consistent with the gene being in an euchromatic state in the approximately 50% of parasites that convert upon Shld1 addition (Fig. 8C-E).”

This statement requires some statistical support from the chipseq data, e.g. diffbind.

Line 420

“Of note, the regions needed to maintain pre-existing HC are different from those capable to nucleate HC de novo”

This is difficult to assess from the presented data. Perhaps a figure showing which bits were tested for nucleation and were effective (F1, F0) and which were required for maintenance (deltaU, deltaU3'-in some subclones) depicted on the same diagram would help. As it stands it seems that delta U3' and F0 and F1 overlap.

Reviewer #3: - Line 191: to control for uneven amplification due to AT bias for D1 and D2 fragments, the authors should use a gDNA titration to determine the primer efficiency and account for this in their calculation (Pfaffl method). This should help to distinguish between technical limitations and true biological variation.

- Fig. S2 is cited as a reference in line 224 – 227 for cloned var and ama1 loci, but the figure only gives information of endogenous HC levels of these genes. I think this is shown in Fig. 2B.

- The experiments integrating sequences of the AP2-G locus into the euchromatic Pf3D7_11444000 gene nicely delineate a sequence at the gene start to be capable of nucleating heterochromatin. However, the sequence is less efficient than parts of the 5’ region of var genes. I was wondering: could the inefficient heterochromatin detection not also be a result of the absence of sequences from area -899 to -130, which were identified as critical for heterochromatin maintenance in Fig. 6 and 7? If I understand correctly, these sequences are missing in the clones carrying F1 or F1+F0.

- Sublcones were generated for U3’, presumably because the original line still contained wt DNA according to Figure S10. As the clonal data for both, nucleation (F1+F0 or F1) and maintenance (U3’) demonstrate a substantial degree of clonal variability, how confident are you that you don’t miss an effect for the other (uncloned) regions?

- Fig. 4F indicates that the heterochromatin boundaries are strictly delimited by the CDS of Pf3D7_1144400. Could the authors speculate what may determine those boundaries in this locus?

- In Fig 4E the H3K9ac pattern is shown in the recombinant locus. What is the H3K9ac/H3K9me3 pattern in WT parasites at this locus? Is this gene normally transcribed and carries promoter acetylations (though the gene is non-essential)?

- In Fig 4E the H3K9ac pattern is shown in the recombinant locus. What is the H3K9ac/H3K9me3 pattern in WT parasites at this locus? Is this gene normally transcribed and carries promoter acetylations (though the gene is non-essential)?

- AP2-G is known to bind to its own promoter and enhance its own transcription. Are there particular sequence motifs evident in the areas for nucleation and maintenance (-899 to -130)? Where are the multiple AP2-G binding sites relative to the identified regions? Are any other predicted TF binding sites or motifs evident? In line 260, the authors mention sequence repetitiveness as a potential factor for making nucleation more effective. Therefore, I think analysis of the primary sequence may help in understanding the results better.

- What is the hypothesis on heterochromatin re-established once the parasites have entered the sexual differentiation program? AP2-G is only briefly unpacked and expressed during commitment and regains a heterochromatic state during gametocyte differentiation. Wouldn’t this argue against the inheritance model of heterochromatin nucleation or do the authors think that other mechanisms are relevant in this context?

- As var genes are grouped into subgroups (A-E) according to their 5’ regions and genomic position that differ significantly in their switching dynamics, it would be meaningful and informative to report what group the analysed variants belong to.

- Please use parasite line names consistently throughout all figures, e.g. I believe the line shown in Fig. 6A as �5’ap2-g (F1+F0 sc A3) is referred to as A3_F1+F0 in Fig. 4 with the heading “ap2-g F1+F0 subclones”. I found this at times hard to follow. A family tree of the clones may be helpful.

- Please report SD, not SEM for all graphs. SEM is an unsuitable statistical value for an experiment with two replicates.

- Please mention why the expected copy number level in Fig. 2D higher (2x) for ama1 and Pf3D7_1240300. Is this because the endogenous sequence has not been deleted?

- Please indicate the location of the TSS of the genes in Fig. 5A, C

- In Fig. 7, I would suggest to highlight the identified “heterochromatin maintenance” region between-899 to -130.

- Fig. 4F indicates that the heterochromatin boundaries are strictly delimited by the CDS of Pf3D7_1144400. Could the authors speculate what may determine those boundaries in this locus?

- What is the hypothesis on heterochromatin re-established once the parasites have entered the sexual differentiation program? AP2-G is only briefly unpacked and expressed during commitment and regains a heterochromatic state during gametocyte differentiation. Wouldn’t this argue against the inheritance model of heterochromatin nucleation or do the authors think that other mechanisms are relevant in this context?

PLOS authors have the option to publish the peer review history of their article (what does this mean? ). If published, this will include your full peer review and any attached files.

**Do you want your identity to be public for this peer review?** For information about this choice, including consent withdrawal, please see our Privacy Policy .

Reviewer #1: No

Reviewer #2: No

Reviewer #3: No

**Figure resubmission:**
---

## [Editor Report · Decision Letter 1]

16 Apr 2025

Dear Dr. Cortes,

We are pleased to inform you that your manuscript 'Heterochromatin de novo formation and maintenance in Plasmodium falciparum' has been provisionally accepted for publication in PLOS Pathogens. 

We were happy to see the extensive modifications and revisions provided in response to the reviewers' suggestions. Given the initial enthusiasm expressed by all three reviewers for the original version of the paper, we are satisfied that the changes have made the manuscript suitable for publication. 

Best regards,

Kirk W. Deitsch

Academic Editor

PLOS Pathogens

Margaret Phillips

Section Editor

PLOS Pathogens

Sumita Bhaduri-McIntosh

Editor-in-Chief

PLOS Pathogens

orcid.org/0000-0003-2946-9497

Michael Malim

Editor-in-Chief

PLOS Pathogens

orcid.org/0000-0002-7699-2064
---

## [Editor Report · Acceptance letter]

Dear Dr. Cortés,

We are delighted to inform you that your manuscript, "Heterochromatin de novo formation and maintenance in Plasmodium falciparum," has been formally accepted for publication in PLOS Pathogens.

Best regards,

Sumita Bhaduri-McIntosh

Editor-in-Chief

PLOS Pathogens

orcid.org/0000-0003-2946-9497

Michael Malim

Editor-in-Chief

PLOS Pathogens

orcid.org/0000-0002-7699-2064